# Out-of-distribution Generalization in the Presence of Nuisance-Induced Spurious Correlations

**Aahlad Puli**[1]*    **Lily H. Zhang**    **Eric K. Oermann**    **Rajesh Ranganath**
New York University

## Abstract

In many prediction problems, spurious correlations are induced by a changing relationship between the label and a nuisance variable that is also correlated with the covariates. For example, in classifying animals in natural images, the background, which is a nuisance, can predict the type of animal. This nuisance-label relationship does not always hold, and the performance of a model trained under one such relationship may be poor on data with a different nuisance-label relationship. To build predictive models that perform well regardless of the nuisance-label relationship, we develop Nuisance-Randomized Distillation (NuRD). We introduce the nuisance-randomized distribution, a distribution where the nuisance and the label are independent. Under this distribution, we define the set of representations such that conditioning on any member, the nuisance and the label remain independent. We prove that the representations in this set always perform better than chance, while representations outside of this set may not. NuRD finds a representation from this set that is most informative of the label under the nuisance-randomized distribution, and we prove that this representation achieves the highest performance regardless of the nuisance-label relationship. We evaluate NuRD on several tasks including chest X-ray classification where, using non-lung patches as the nuisance, NuRD produces models that predict pneumonia under strong spurious correlations.

## 1 Introduction

Spurious correlations are relationships between the label and the covariates that are prone to change between training and test distributions (Gulrajani and Lopez-Paz, 2020). Predictive models that exploit spurious correlations can perform worse than even predicting without covariates on the test distribution (Arjovsky et al., 2019). Discovering spurious correlations requires more than the training distribution because any single distribution has a fixed label-covariate relationship. Often, spurious correlations are discovered by noticing different relationships across multiple distributions between the label and *nuisance* factors correlated with the covariates. We call these *nuisance-induced spurious correlations*.

For example, in classifying cows vs. penguins, typical images have cows appear on grasslands and penguins appear near snow, their respective natural habitats (Arjovsky et al., 2019; Geirhos et al., 2020), but these animals can be photographed outside their habitats. In classifying hair color from celebrity faces on CelebA (Liu et al., 2015), gender is correlated with the hair color. This relationship may not hold in different countries (Sagawa et al., 2020). In language, sentiment of a movie review determines the types of words used in the review to convey attitudes and opinions. However, directors' names appear in the reviews and are correlated with positive sentiment in time periods where directors make movies that are well-liked (Wang and Culotta, 2020a). In X-ray classification, conditions like pneumonia are spuriously correlated with non-physiological traits of X-ray images due to the association between the label and hospital X-ray collection protocols (Zech et al., 2018). Such factors are rarely recorded in datasets but produce subtle differences in X-ray images that convolutional networks easily learn (Badgeley et al., 2019).

We formalize nuisance-induced spurious correlations in a nuisance-varying family of distributions where any two distributions are different only due to the differences in the nuisance-label relationship. As the nuisance is informative of the label, predictive models exploit the nuisance-label relationship to achieve the best performance on any single member of the family. However, predictive models that perform best on one member can perform even worse than predicting without any covariates on another member, which may be out-of-distribution (OOD). We develop NuRD to use

---

*[1]Corresponding email: aahlad@nyu.edu. The code is available here.

data collected under one nuisance-label relationship to build predictive models that perform well on other members of the family regardless of the nuisance-label relationship in that member.

In section 2, we motivate and develop ideas that help guarantee performance on every member of the family. The first is the *nuisance-randomized distribution*: a distribution where the nuisance is independent of the label. An example is the distribution where cows and penguins have equal chances of appearing on backgrounds of grass or snow. The second is an *uncorrelating representation*: a representation of the covariates such that under the nuisance-randomized distribution, the nuisance remains independent of the label after conditioning on the representation. The set of such representations is the *uncorrelating set*. We show that the nuisance-randomized conditional of the label given an uncorrelating representation has performance guarantees: such conditionals perform as well or better than predicting without covariates on every member in the family while other conditionals may not. Within the uncorrelating set, we characterize one that is optimal on every member of the nuisance-varying family *simultaneously*. We then prove that the same optimal performance can be realized by uncorrelating representations that are most informative of the label under the nuisance-randomized distribution.

Following the insights in section 2, we develop Nuisance-Randomized Distillation (NURD) in section 3. NURD finds an uncorrelating representation that is maximally informative of the label under the nuisance-randomized distribution. NURD's first step, *nuisance-randomization*, breaks the nuisance-label dependence to produce nuisance-randomized data. We provide two nuisance randomization methods based on generative models and reweighting. The second step, *distillation*, maximizes the information a representation has with the label on the nuisance-randomized data over the uncorrelating set. We evaluate NURD on class-conditional Gaussians, labeling colored MNIST images (Arjovsky et al., 2019), distinguishing waterbirds from landbirds, and classifying chest X-rays. In the latter, using the non-lung patches as the nuisance, NURD produces models that predict pneumonia under strong spurious correlations.

## 2    NUISANCE-RANDOMIZATION AND UNCORRELATING SETS

We formalize nuisance-induced spurious correlations via a family of data generating processes. Let $\mathbf{y}$ be the label, $\mathbf{z}$ be the nuisance, and $\mathbf{x}$ be the covariates (i.e. features). The family consists of distributions where the only difference in the members of the family comes from the difference in their nuisance-label relationships. Let $D$ index a family of distributions $\mathcal{F} = \{p_D\}_D$; a member $p_D$ in the nuisance-varying family of distributions $\mathcal{F}$ takes the following form:

$$p_D(\mathbf{y}, \mathbf{z}, \mathbf{x}) = p(\mathbf{y})p_D(\mathbf{z} \mid \mathbf{y})p(\mathbf{x} \mid \mathbf{z}, \mathbf{y}), \tag{1}$$

where $p_D(\mathbf{z} \mid \mathbf{y})$ is positive and bounded for any $\mathbf{y}$ where $p(\mathbf{y}) > 0$ and any $\mathbf{z}$ in the family's nuisance space $S_{\mathcal{F}}$. This family is called the nuisance-varying family. Due to changing nuisance-label relationships in this family, the conditional distribution of the label $\mathbf{y}$ given the covariates $\mathbf{x}$ in one member, e.g. the training distribution, can perform worse than predicting without covariates on another member of the family, e.g. a test distribution with a different nuisance-label relationship. We define performance of a model $\hat{p}(\mathbf{y} \mid \mathbf{x})$ on a distribution $p_{te}$ as the negative expected KL-divergence from the true conditional $p_{te}(\mathbf{y} \mid \mathbf{x})$: $\texttt{Perf}_{p_{te}}(\hat{p}(\mathbf{y} \mid \mathbf{x})) = -\mathbb{E}_{p_{te}(\mathbf{x})}\text{KL}\left[p_{te}(\mathbf{y} \mid \mathbf{x}) \parallel \hat{p}(\mathbf{y} \mid \mathbf{x})\right]$. Higher is better. This performance equals the expected log-likelihood up to a constant, $C_{p_{te}} = \mathbf{H}_{p_{te}}(\mathbf{y} \mid \mathbf{x})$, that only depends on the $p_{te}$: $\texttt{Perf}_{p_{te}}(\hat{p}(\mathbf{y} \mid \mathbf{x})) = \mathbb{E}_{p_{te}(\mathbf{y}, \mathbf{x})} \log \hat{p}(\mathbf{y} \mid \mathbf{x}) + C_{p_{te}}$. Consider the following example family $\{q_a\}_{a \in \mathbb{R}}$:

$$\mathbf{y} \sim \mathcal{N}(0, 1) \quad \mathbf{z} \sim \mathcal{N}(a\mathbf{y}, 0.5) \quad \mathbf{x} = [\mathbf{x}_1 \sim \mathcal{N}(\mathbf{y} - \mathbf{z}, 1.5), \mathbf{x}_2 \sim \mathcal{N}(\mathbf{y} + \mathbf{z}, 0.5)]. \tag{2}$$

Given training distribution $p_{tr} = q_1$ and test distribution $p_{te} = q_{-1}$, the conditional $p_{tr}(\mathbf{y} \mid \mathbf{x})$ performs even worse than predicting without covariates, $\texttt{Perf}_{p_{te}}(p(\mathbf{y})) \geq \texttt{Perf}_{p_{te}}(p_{tr}(\mathbf{y} \mid \mathbf{x}))$; see appendix A.9 for the proof. The problem is that $p_{tr}(\mathbf{y} \mid \mathbf{x})$ utilizes label-covariate relationships that do not hold when the nuisance-label relationships change. When the changing nuisance-label relationship makes the conditional $p_D(\mathbf{y} \mid \mathbf{x})$ of one member unsuitable for another $p'_D \in \mathcal{F}$, the family exhibits *nuisance-induced spurious correlations*.

Next, we identify a conditional distribution with performance guarantees across all members of the family. We develop two concepts to guarantee performance on every member of the nuisance-varying family: the nuisance-randomized distribution and uncorrelating representations.

**Definition 1.** *The **nuisance-randomized distribution** is $p_{\perp}(\mathbf{x}, \mathbf{y}, \mathbf{z}) = p(\mathbf{x} \mid \mathbf{y}, \mathbf{z})p_{tr}(\mathbf{z})p(\mathbf{y})$.*[1]

---

[1] Different marginal distributions $p_{\perp}(\mathbf{z})$ produce different distributions where the label and nuisance are independent. The results are insensitive to the choice as long as $p_{\perp}(\mathbf{z}) > 0$ for any $\mathbf{z} \in S_{\mathcal{F}}$. One distribution that satisfies this requirement is $p_{\perp}(\mathbf{z}) = p_{tr}(\mathbf{z})$. See lemma 2.

In the cows vs. penguins example, $p_{\perp}$ is the distribution where either animal has an equal chance to appear on backgrounds of grass or snow. The motivation behind the nuisance-randomized distribution is that when the nuisance is independent of the label, (noisy[2]) functions of only the nuisance are not predictive of the label. If the covariates only consist of (noisy) functions of either the nuisance or the label but never a mix of the two (an example of mixing is $\mathbf{x}_1 = \mathbf{y} - \mathbf{z} + noise$), then the conditional $p_{\perp}(\mathbf{y} \mid \mathbf{x})$ does not vary with the parts of $\mathbf{x}$ that are (noisy) functions of just the nuisance. Thus, $p_{\perp}(\mathbf{y} \mid \mathbf{x})$ ignores the features which have changing relationships with the label.

How about nuisance-varying families where the covariates contain functions that mix the label and the nuisance? Equation (2) is one such family, where the covariates $\mathbf{x}_1$ and $\mathbf{x}_2$ are functions of both the label and the nuisance. In such nuisance-varying families, the conditional $p_{\perp}(\mathbf{y} \mid \mathbf{x})$ can use functions that mix the label and the nuisance even though the nuisance is not predictive of the label by itself. These mixed functions have relationships with the label which change across the family; for example in eq. (2), the coordinate $\mathbf{x}_1$ is correlated positively with the label under $q_0$ but negatively under $q_{-2}$. Then, under changes in the nuisance-label relationship, the conditional $p_{\perp}(\mathbf{y} \mid \mathbf{x})$ can perform worse than predicting without covariates because it utilizes a relationship, via these mixed features, that no longer holds. See appendix A.9 for details.

We address this performance degradation of $p_{\perp}(\mathbf{y} \mid \mathbf{x})$ by introducing representations that help avoid reliance on functions that mix the label and the nuisance. We note that when the conditional $p_{\perp}(\mathbf{y} \mid \mathbf{x})$ uses functions that mix the label and the nuisance, knowing the exact value of the nuisance should improve the prediction of the label, i.e. $\mathbf{y} \not\perp_{p_{\perp}} \mathbf{z} \mid \mathbf{x}$. Therefore, to avoid reliance on mixed functions, we define *uncorrelating* representations $r(\mathbf{x})$, where the nuisance does not provide any extra information about the label given the representation:

**Definition 2.** *An uncorrelating set of representations is* $\mathcal{R}(p_{\perp})$ *s.t.* $\forall r \in \mathcal{R}(p_{\perp}), \quad \mathbf{y} \perp\!\!\!\perp_{p_{\perp}} \mathbf{z} \mid r(\mathbf{x})$.

In the example in eq. (2), $r(\mathbf{x}) = \mathbf{x}_1 + \mathbf{x}_2$ is an uncorrelating representation because it is purely a function of the label and the noise. Conditional distributions $p_{\perp}(\mathbf{y} \mid r(\mathbf{x}))$ for any uncorrelating $r(\mathbf{x})$ only depend on properties that are shared across all distributions in the nuisance-varying family. Specifically, for $r \in \mathcal{R}(p_{\perp})$, the conditional distribution $p_{\perp}(\mathbf{y} \mid r(\mathbf{x}))$ uses $p(r(\mathbf{x}) \mid \mathbf{y}, \mathbf{z})$ and $p(\mathbf{y})$ which are both shared across all members of the family $\mathcal{F}$. For $\mathbf{z}'$ such that $p_{\perp}(\mathbf{z}' \mid r(\mathbf{x})) > 0$,

$$p_{\perp}(\mathbf{y} \mid r(\mathbf{x})) = p_{\perp}(\mathbf{y} \mid r(\mathbf{x}), \mathbf{z}') = \frac{p_{\perp}(\mathbf{y} \mid \mathbf{z}')p_{\perp}(r(\mathbf{x}) \mid \mathbf{y}, \mathbf{z}')}{p_{\perp}(r(\mathbf{x}) \mid \mathbf{z}')} = \frac{p(\mathbf{y})p(r(\mathbf{x}) \mid \mathbf{y}, \mathbf{z}')}{\mathbb{E}_{p(\mathbf{y})}p(r(\mathbf{x}) \mid \mathbf{y}, \mathbf{z}')}. \quad (3)$$

This fact helps characterize the performance of $p_{\perp}(\mathbf{y} \mid r(\mathbf{x}))$ on any member $p_{te} \in \mathcal{F}$:

$$\texttt{Perf}_{p_{te}}(p_{\perp}(\mathbf{y} \mid r(\mathbf{x}))) = \texttt{Perf}_{p_{te}}(p(\mathbf{y})) + \mathbb{E}_{p_{te}(\mathbf{y}, \mathbf{z})} \texttt{KL}\left[p(r(\mathbf{x}) \mid \mathbf{y}, \mathbf{z}) \parallel \mathbb{E}_{p(\mathbf{y})}p(r(\mathbf{x}) \mid \mathbf{y}, \mathbf{z})\right]. \quad (4)$$

As KL-divergence is non-negative, for any uncorrelating representation $r$, the conditional $p_{\perp}(\mathbf{y} \mid r(\mathbf{x}))$ does at least as well as predicting without covariates for all members $p_{te} \in \mathcal{F}$: $\texttt{Perf}_{p_{te}}(p_{\perp}(\mathbf{y} \mid r(\mathbf{x}))) \geq \texttt{Perf}_{p_{te}}(p(\mathbf{y}))$. See appendix A.3 for the formal derivation. In fact, we show in appendix A.5 that when the identity representation $r(\mathbf{x}) = \mathbf{x}$ is uncorrelating, then $p_{\perp}(\mathbf{y} \mid \mathbf{x})$ is minimax optimal for a family with sufficiently diverse nuisance-label relationships.

Equation (4) lower bounds the performance of $p_{\perp}(\mathbf{y} \mid r(\mathbf{x}))$ for any representation in the uncorrelating set across all $p_{te} \in \mathcal{F}$. However, it does not specify which of these representations leads to the best performing conditional. For example, between two uncorrelating representations like the shape of the animal and whether the animal has horns, which predicts better? Next, we characterize uncorrelating representations that are *simultaneously* optimal for all test distributions $p_{te} \in \mathcal{F}$.

**Optimal uncorrelating representations.** As we focus on nuisance-randomized conditionals, henceforth, by performance of $r(\mathbf{x})$, we mean the performance of $p_{\perp}(\mathbf{y} \mid r(\mathbf{x}))$: $\texttt{Perf}_{p_{te}}(r(\mathbf{x})) = \texttt{Perf}_{p_{te}}(p_{\perp}(\mathbf{y} \mid r(\mathbf{x})))$. Consider two uncorrelating representations $r, r_2$, where the pair $(r, r_2)$ is also uncorrelating. How can $r_2(\mathbf{x})$ dominate $r(\mathbf{x})$ in performance across the nuisance-varying family? Equation (3) shows that

$$p_{\perp}(\mathbf{y} \mid [r(\mathbf{x}), r_2(\mathbf{x})]) \propto p(\mathbf{y})p(r(\mathbf{x}) \mid r_2(\mathbf{x}), \mathbf{y}, \mathbf{z} = \mathbf{z})p(r_2(\mathbf{x}) \mid \mathbf{y}, \mathbf{z} = \mathbf{z}).$$

If $r_2(\mathbf{x})$ **blocks** the dependence between the label and $r(\mathbf{x})$, i.e. $r(\mathbf{x}) \perp\!\!\!\perp_{p_{\perp}} \mathbf{y} \mid r_2(\mathbf{x}), \mathbf{z}$, then knowing $r$ does not change the performance when $r_2$ is known, suggesting that blocking relates to performance. In theorem 1, we show that the *maximally blocking* uncorrelating representation is simultaneously optimal: its performance is as good or better than every other uncorrelating representation on every distribution in the nuisance-varying family. We state the theorem first:

---

[2]Noisy functions of a variable are functions of that variable and exogenous noise.

**Theorem 1.** *Let $r^* \in \mathcal{R}(p_\perp)$ be **maximally blocking**: $\forall r \in \mathcal{R}(p_\perp), \quad \mathbf{y} \perp\!\!\!\perp_{p_\perp} r(\mathbf{x}) \mid \mathbf{z}, r^*(\mathbf{x})$. Then,*

1. *($Simultaneous\ optimality$) $\forall p_{te} \in \mathcal{F}, \forall r \in \mathcal{R}(p_\perp), \quad \mathtt{Perf}_{p_{te}}(r^*(\mathbf{x})) \geq \mathtt{Perf}_{p_{te}}(r(\mathbf{x}))$.*

2. *($Information\ maximality$) $\forall r(\mathbf{x}) \in \mathcal{R}(p_\perp), \quad \mathbf{I}_{p_\perp}(\mathbf{y}; r^*(\mathbf{x})) \geq \mathbf{I}_{p_\perp}(\mathbf{y}; r(\mathbf{x}))$.*

3. *($Information\ maximality\ implies\ simultaneous\ optimality$) $\forall r' \in \mathcal{R}(p_\perp)$,*

$$\mathbf{I}_{p_\perp}(\mathbf{y}; r'(\mathbf{x})) = \mathbf{I}_{p_\perp}(\mathbf{y}; r^*(\mathbf{x})) \quad \implies \quad \forall p_{te} \in \mathcal{F}, \quad \mathtt{Perf}_{p_{te}}(r^*(\mathbf{x})) = \mathtt{Perf}_{p_{te}}(r'(\mathbf{x})).$$

The proof is in appendix A.4. The first part of theorem 1, *simultaneous optimality*, says that a maximally blocking uncorrelating representation $r^*$ dominates every other $r \in \mathcal{R}(p_\perp)$ in performance on *every* test distribution in the family. In the cows vs. penguins example, the segmented foreground that contains only the animal is a maximally blocking representation because the animal blocks the dependence between the label and any other semantic feature of the animal.

The second and third parts of theorem 1 are useful for algorithm building. The second part proves that a maximally blocking $r^*$ is also maximally informative of the label under $p_\perp$, indicating how to find a simultaneously optimal uncorrelating representation. What about other information-maximal uncorrelating representations? The third part shows that if an uncorrelating representation $r'$ has the same mutual information with the label (under $p_\perp$) as the maximally blocking $r^*$, then $r'$ achieves the same simultaneously optimal performance as $r^*$. In the cows vs. penguins example, an example of a maximally informative uncorrelating representation is the number of legs of the animal because the rest of the body does not give more information about the label. The second and third parts of theorem 1 together show that finding an uncorrelating representation that maximizes information under the nuisance-randomized distribution finds a simultaneously optimal uncorrelating $r(\mathbf{x})$.

## 3    NUISANCE-RANDOMIZED DISTILLATION (NURD)

Theorem 1 says a representation that maximizes information with the label under the nuisance-randomized distribution has the best performance within the uncorrelating set. We develop a representation learning algorithm to maximize the mutual information between the label and a representation in the uncorrelating set under the nuisance-randomized distribution. We call this algorithm Nuisance-Randomized Distillation (NURD). NURD has two steps. The first step, called nuisance randomization, creates an estimate of the nuisance-randomized distribution. The second step, called distillation, finds a representation in the uncorrelating set with the maximum information with the label under the estimate of the nuisance-randomized distribution from step one.

**Nuisance Randomization.** We estimate the nuisance-randomized distribution with generative models or by reweighting existing data. Generative-NURD uses the fact that $p(\mathbf{x} \mid \mathbf{y}, \mathbf{z})$ is the same for each member of the nuisance-varying family $\mathcal{F}$. With an estimate of this conditional denoted $\hat{p}(\mathbf{x} \mid \mathbf{y}, \mathbf{z})$, generative-NURD's estimate of the nuisance-randomized distribution is $\mathbf{z} \sim p_{tr}(\mathbf{z}), \mathbf{y} \sim p(\mathbf{y}), \mathbf{x} \sim \hat{p}(\mathbf{x} \mid \mathbf{y}, \mathbf{z})$. For high dimensional $\mathbf{x}$, the estimate $\hat{p}(\mathbf{x} \mid \mathbf{y}, \mathbf{z})$ can be constructed with deep generative models. Reweighting-NURD importance weights the data from $p_{tr}$ by $p(\mathbf{y})/p_{tr}(\mathbf{y} \mid \mathbf{z})$, making it match the nuisance-randomized distribution:

$$p_\perp(\mathbf{x}, \mathbf{y}, \mathbf{z}) = p(\mathbf{y})p_{tr}(\mathbf{z})p(\mathbf{x} \mid \mathbf{y}, \mathbf{z}) = p(\mathbf{y})p_{tr}(\mathbf{z})\frac{p_{tr}(\mathbf{y} \mid \mathbf{z})}{p_{tr}(\mathbf{y} \mid \mathbf{z})}p(\mathbf{x} \mid \mathbf{y}, \mathbf{z}) = \frac{p(\mathbf{y})}{p_{tr}(\mathbf{y} \mid \mathbf{z})}p_{tr}(\mathbf{x}, \mathbf{y}, \mathbf{z}).$$

Reweighting-NURD uses a model trained on samples from $p_{tr}$ to estimate $\frac{p(\mathbf{y})}{p_{tr}(\mathbf{y} \mid \mathbf{z})}$.

**Distillation.** Distillation seeks to find the representation in the uncorrelating set that maximizes the information with the label under $\hat{p}_\perp$, the estimate of the nuisance-randomized distribution. Maximizing the information translates to maximizing likelihood because the entropy $\mathbf{H}_{\hat{p}_\perp}(\mathbf{y})$ is constant with respect to the representation $r_\gamma$ parameterized by $\gamma$:

$$\mathbf{I}_{\hat{p}_\perp}(r_\gamma(\mathbf{x}); \mathbf{y}) - \mathbf{H}_{\hat{p}_\perp}(\mathbf{y}) = \mathbb{E}_{\hat{p}_\perp(\mathbf{y}, r_\gamma(\mathbf{x}))} \log \hat{p}_\perp(\mathbf{y} \mid r_\gamma(\mathbf{x})) = \max_\theta \mathbb{E}_{\hat{p}_\perp(\mathbf{y}, r_\gamma(\mathbf{x}))} \log p_\theta(\mathbf{y} \mid r_\gamma(\mathbf{x})).$$

Theorem 1 requires the representations be in the uncorrelating set. When conditioning on representations in the uncorrelating set, the nuisance has zero mutual information with the label: $\mathbf{I}_{p_\perp}(\mathbf{y}; \mathbf{z} \mid r_\gamma(\mathbf{x})) = 0$. We operationalize this constraint by adding a conditional mutual information penalty to the maximum likelihood objective with a tunable scalar parameter $\lambda$

$$\max_{\theta, \gamma} \mathbb{E}_{\hat{p}_\perp(\mathbf{y}, \mathbf{z}, \mathbf{x})} \log p_\theta(\mathbf{y} \mid r_\gamma(\mathbf{x})) - \lambda \mathbf{I}_{\hat{p}_\perp}(\mathbf{y}; \mathbf{z} \mid r_\gamma(\mathbf{x})). \tag{5}$$

The objective in eq. (5) can have local optima when the representation is a function of the nuisance and exogenous noise (noise that generates the covariates given the nuisance and the label). The intuition behind these local optima is that the value of introducing information that predicts the label does not exceed the cost of the introduced conditional dependence. Appendix A.6 gives a formal discussion and an example with such local optima. Annealing $\lambda$, which controls the cost of conditional dependence, can mitigate the local optima issue at the cost of setting annealing schedules.

Instead, we restrict the distillation step to search over representations $r_\gamma(\mathbf{x})$ that are also marginally independent of the nuisance $\mathbf{z}$ under $p_\perp$, i.e. $\mathbf{z} \perp\!\!\!\perp_{p_\perp} r_\gamma(\mathbf{x})$. This additional independence removes representations that depend on the nuisance but are not predictive of the label; in turn, this removes local optima that correspond to functions of the nuisance and exogenous noise. In the cows vs. penguins example, representations that are functions of the background only, like the presence of snow, are uncorrelating but do not satisfy the marginal independence. Together, the conditional independence $\mathbf{y} \perp\!\!\!\perp_{p_\perp} \mathbf{z} \mid r_\gamma(\mathbf{x})$ and marginal independence $\mathbf{z} \perp\!\!\!\perp_{p_\perp} r_\gamma(\mathbf{x})$ hold if and only if the representation and the label are jointly independent of the nuisance : $(\mathbf{y}, r_\gamma(\mathbf{x})) \perp\!\!\!\perp_{p_\perp} \mathbf{z}$. Using mutual information to penalize joint *dependence* (instead of the penalty in eq. (5)), the distillation step in NURD is

$$\max_{\theta,\gamma} \mathbb{E}_{\hat{p}_\perp(\mathbf{y},\mathbf{z},\mathbf{x})} \log p_\theta(\mathbf{y} \mid r_\gamma(\mathbf{x})) - \lambda \mathbf{I}_{\hat{p}_\perp}([\mathbf{y}, r_\gamma(\mathbf{x})]; \mathbf{z}). \tag{6}$$

We show in lemma 4 that within the set of representations that satisfy joint independence, NURD learns a representation that is simultaneously optimal in performance on all members of the nuisance-varying family. To learn representations using gradients, the mutual information needs to be estimated in a way that is amenable to gradient optimization. To achieve this, we estimate the mutual information in NURD via the classification-based density-ratio estimation trick (Sugiyama et al., 2012). We use a *critic model* $p_\phi$ to estimate said density ratio. We describe this technique in appendix A.1 for completeness. We implement the distillation step as a bi-level optimization where the outer loop optimizes the predictive model $p_\theta(\mathbf{y} \mid r_\gamma(\mathbf{x}))$ and the inner loop optimizes the critic model $p_\phi$ which helps estimate the mutual information.

**Algorithm.** We give the full algorithm boxes for both reweighting-NURD and generative-NURD in appendix A.1. In reweighting-NURD, to avoid poor weight estimation due to models memorizing the training data, we use cross-fitting; see algorithm 1. The setup of nuisance-induced spurious correlations in eq. (1) assumes $p(\mathbf{y})$ is fixed across distributions within the nuisance-varying family $\mathcal{F}$. This condition can be relaxed when $p_{te}(\mathbf{y})$ is known; see appendix A.1.

## 4 RELATED WORK

In table 1, we summarize key differences between NURD and the related work: invariant learning (Arjovsky et al., 2019; Krueger et al., 2020), distribution matching (Mahajan et al., 2020; Guo et al., 2021), shift-stable prediction (Subbaswamy et al., 2019a), group-DRO (Sagawa et al., 2019), and causal regularization (Veitch et al., 2021; Makar et al., 2021). We detail the differences here.

**Nuisance versus Environment.** In general, an environment is a distribution with a specific spurious correlation (Sagawa et al., 2019). When the training and test distributions are members of the same nuisance-varying family, environments denote specific nuisance-label relationships. In contrast, nuisances are variables whose changing relationship with the label induces spurious correlations. While obtaining data from diverse environments requires data collection from sufficiently different sources, one can specify nuisances from a single source of data via domain knowledge.

**Domain generalization, domain-invariant learning, and subgroup robustness** We briefly mention existing methods that aim to generalize to unseen test data and focus on how these methods can suffer in the presence of nuisance-induced spurious correlations; for a more detailed presentation, see appendix A.2. Domain generalization and domain-invariant learning methods assume the training data consists of multiple *sufficiently different* environments to generalize to unseen test data that is related to the given environments or subgroups (Li et al., 2018a; Arjovsky et al., 2019; Akuzawa et al., 2019; Mahajan et al., 2020; Krueger et al., 2020; Bellot and van der Schaar, 2020; Li et al., 2018b; Goel et al., 2020; Wald et al., 2021; Ganin et al., 2016; Xie et al., 2017; Zhang et al., 2018; Ghimire et al., 2020; Adeli et al., 2021; Zhang et al., 2021). Due to its focus on nuisances, NURD works with data from a single environment. Taking a distributional robustness (Duchi et al., 2021) approach, Sagawa et al. (2019) used group-DRO to build models that perform well on every one of a finite set of known subgroups. Other work also aims to minimize worst subgroup error with a finite number of fixed but unknown subgroups (Lahoti et al., 2020; Martinez et al., 2021); as subgroups

**Table 1:** NuRD vs. methods that use nuisances or environments. In this work, the training data comes from a single member of the family $\mathcal{F}$, i.e. a single environment. For methods that require multiple environments, values of the nuisance can be treated as environment labels. Unlike existing methods, NuRD works with high-dimensional nuisances without requiring them at test time.

|  | Invariant | Dist. match | Shift-stable | Group-DRO | Causal reg. | NuRD |
|---|---|---|---|---|---|---|
| High-dim $\mathbf{z}$ | ✗ | ✗ | ✓ | ✗ | ✗ | ✓ |
| No test-time $\mathbf{z}$ | ✓ | ✓ | ✗ | ✓ | ✓ | ✓ |

are unknown, they only find an approximate minimizer of the worst subgroup error in general even with infinite data. While these methods (Lahoti et al., 2020; Martinez et al., 2021) were developed to enforce fairness with respect to a sensitive attribute, they can be applied to OOD generalization with the nuisance treated as the sensitive attribute; see (Creager et al., 2021). Given the nuisance, existence of a finite number of subgroups maps to an additional discreteness assumption on the nuisance variable; in contrast, NuRD works with general nuisances. Given a high dimensional $\mathbf{z}$, as in our experiments, defining groups based on the value of the nuisance like in (Sagawa et al., 2019) typically results in groups with at most one sample; with the resulting groups, methods that minimize worst subgroup error will encourage memorizing the training data.

**Nuisance as the environment label for domain generalization.** Domain generalization methods are inapplicable when the training data consists only of a single environment. In this work, the training data comes from only one member of the nuisance-varying family, i.e. from a single environment. What if one treats groups defined by nuisance values as environments? Using the nuisance as the environment label can produce non-overlapping supports (over the covariates) between environments. In Colored-MNIST for example, splitting based on color produces an environment of green images and an environment of red images. When the covariates do not overlap between environments, methods such as Arjovsky et al. (2019); Krueger et al. (2020) will not produce invariant representations because the model can segment out the covariate space and learn separate functions for each environment. Methods based on conditional distribution matching (Mahajan et al., 2020; Guo et al., 2021) build representations that are conditionally independent of the environment variable given the label. When the training data is split into groups based on the nuisance value, representations built by these methods are independent of the nuisance given the label. However, splitting on a high-dimensional nuisance like image patches tends to yield many groups with only a single image. Matching distributions of representations for the same label across all environments is not possible when some environments only have one label.

**Causal learning and shift-stable prediction.** Anticausal learning (Schölkopf et al., 2012) assumes a causal generative process for a class of distributions like the nuisance-varying family in eq. (1). In such an interpretation, the label $\mathbf{y}$ and nuisance $\mathbf{z}$ cause the image $\mathbf{x}$, and under independence of cause and mechanism, $p(\mathbf{x} \mid \mathbf{y}, \mathbf{z})$ is fixed — in other words, independent — regardless of the distribution $p_D(\mathbf{y}, \mathbf{z})$. A closely related idea to NuRD is that of Shift-Stable Prediction (Subbaswamy et al., 2019a;b). Subbaswamy et al. (2019a) perform *graph surgery* to learn $p_{\perp}(\mathbf{y} \mid \mathbf{x}, \mathbf{z})$ assuming access to $\mathbf{z}$ during test time. Shift-stable models are not applicable without nuisances at test time while NuRD only requires nuisances during training. However, if the nuisance is available at test time, the combined covariate-set $[\mathbf{x}, \mathbf{z}]$ is 1) uncorrelating because $\mathbf{y} \perp\!\!\!\perp_{p_{\perp}} \mathbf{z} \mid [\mathbf{x}, \mathbf{z}]$ and 2) maximally blocking because $r([\mathbf{x}, \mathbf{z}]) \perp\!\!\!\perp \mathbf{y} \mid \mathbf{z}, [\mathbf{x}, \mathbf{z}]$, and theorem 1 says $p_{\perp}(\mathbf{y} \mid [\mathbf{x}, \mathbf{z}])$ is optimal.

Two concurrent works also build models using the idea of a nuisance variable. Makar et al. (2021) assume that there exists a stochastic function of $\mathbf{y}$ but not $\mathbf{z}$, called $\mathbf{x}^*$, such that $\mathbf{y} \perp\!\!\!\perp_{p_D} \mathbf{x} \mid \mathbf{x}^*, \mathbf{z}$; they use a marginal independence penalty $r(\mathbf{x}) \perp\!\!\!\perp_{p_{\perp}} \mathbf{z}$. Veitch et al. (2021) use counterfactual invariance to derive a conditional independence penalty $r(\mathbf{x}) \perp\!\!\!\perp_{p_{tr}} \mathbf{z} \mid \mathbf{y}$. The theory in these works requires the nuisance to be discrete, and their algorithms require the nuisance to be both discrete and low-cardinality; NuRD and its theory work with general high-dimensional nuisances. Counterfactual invariance promises that a representation will not vary with the nuisance but it does not produce optimal models in general because it rejects models that depend on functions of the nuisance. On the other hand, the uncorrelating property allows using functions of only the nuisance that are in the covariates to extract further information about the label from rest of the covariates; this leads to better performance in some nuisance-varying families, as we show using the theory of minimal sufficient statistics (Lehmann and Casella, 2006) in appendix A.7.

## 5 EXPERIMENTS

We evaluate the implementations of NURD on class-conditional Gaussians, Colored-MNIST (Arjovsky et al., 2019), Waterbirds (Sagawa et al., 2019), and chest X-rays (Irvin et al., 2019; Johnson et al., 2019). See appendix B for implementation details and further evaluations of NURD.

**Model selection, baselines, and metrics.** Models in both steps of NURD are selected using held-out subsets of the training data. We split the training data into training and validation datasets with an $80 - 20$ split. For nuisance-randomization, this selection uses standard measures of held-out performance. Selection in the distillation step picks models that give the best value of the distillation objective on a held out subset of the *nuisance-randomized* data from NURD's first step.

We compare against Empirical Risk minimization (ERM) because, as discussed in section 4, existing methods that aim to generalize under spurious correlations require assumptions, such as access to multiple environments or discrete nuisance of small cardinality, that do not hold in the experiments. When possible, we report the oracle accuracy or build gold standard models using data that does not have a nuisance-label relationship to exploit. For every method, we report the average accuracy and standard error across 10 runs each with a different random seed. We report the accuracy of each model for each experiment on the test data ($p_{te}$) and on heldout subsets of the original training data ($p_{tr}$) and the estimate of the nuisance-randomized distribution ($\hat{p}_\perp$). For all experiments, we use $\lambda = 1$ and one or two epochs of critic model updates for every predictive model update.

### 5.1 CLASS-CONDITIONAL GAUSSIANS

We generate data as follows: with $\mathcal{B}(0.5)$ as the uniform Bernoulli distribution, $q_a(\mathbf{y}, \mathbf{z}, \mathbf{x})$ is

$$\mathbf{y} \sim \mathcal{B}(0.5) \quad \mathbf{z} \sim \mathcal{N}(a(2\mathbf{y}-1), 1) \quad \mathbf{x} = [\mathbf{x}_1 \sim \mathcal{N}(\mathbf{y}-\mathbf{z}, 9), \mathbf{x}_2 \sim \mathcal{N}(\mathbf{y}+\mathbf{z}, 0.01)]. \quad (7)$$

The training and test sets consist of 10000 samples from $p_{tr} = q_{0.5}$ and 2000 samples from $p_{te} = q_{-0.9}$ respectively. All models in both NURD methods are parameterized with neural networks.

**Table 2:** Accuracy of NURD versus ERM on class conditional Gaussians.

| Method | Heldout $p_{tr}$ | Heldout $\hat{p}_\perp$ | $p_{te}$ |
|---|---|---|---|
| ERM | $84 \pm 0\%$ | $-$ | $39 \pm 0\%$ |
| generative-NURD | $71 \pm 0\%$ | $67 \pm 0\%$ | $58 \pm 0\%$ |
| reweighting-NURD | $71 \pm 1\%$ | $66 \pm 0\%$ | $58 \pm 0\%$ |

**Results.** Table 2 reports results. The test accuracy of predicting with the optimal linear uncorrelating representation $r^*(\mathbf{x}) = \mathbf{x}_1 + \mathbf{x}_2$, is $62\%$; appendix B gives the optimality proof. Both generative-NURD and reweighting-NURD achieve close to this accuracy.

### 5.2 COLORED-MNIST

We construct a colored-MNIST dataset (Arjovsky et al., 2019; Gulrajani and Lopez-Paz, 2020) with images of 0s and 1s. In this dataset, the values in each channel for every pixel are either 0 or 1. We construct two environments and use one as the training distribution and the other as the test. In training, $90\%$ of red images have label 0; $90\%$ of green images have label 1. In test, the relationship is flipped: $90\%$ of the 0s are green, and $90\%$ of the 1s are red. In both training and test, the digit determines the label only $75\%$ of the time, meaning that exploiting the nuisance-label relationship produces better training accuracies. The training and test data consist of 4851 and 4945 samples respectively. We run NURD with the most intense pixel as the nuisance.

**Table 3:** Accuracy of NURD versus ERM on Colored-MNIST. The oracle accuracy is $75\%$.

| Method | Heldout $p_{tr}$ | Heldout $\hat{p}_\perp$ | $p_{te}$ |
|---|---|---|---|
| ERM | $90 \pm 0\%$ | $-$ | $10 \pm 0\%$ |
| generative-NURD | $73 \pm 1\%$ | $80 \pm 1\%$ | $68 \pm 2\%$ |
| reweighting-NURD | $75 \pm 0\%$ | $74 \pm 0\%$ | $75 \pm 0\%$ |

**Results.**  See table 3 for the results. ERM learns to use color, evidenced by the fact that it achieves a test accuracy of only $10\%$. The oracle accuracy of $75\%$ is the highest achievable by models that do not use color because the digit only predicts the label with $75\%$ accuracy. While generative-NuRD has an average accuracy close to the oracle, reweighting-NuRD matches the oracle at $75\%$.

## 5.3  LEARNING TO CLASSIFY WATERBIRDS AND LANDBIRDS

Sagawa et al. (2019) consider the task of detecting the type of bird (water or land) from images where the background is a nuisance. Unlike Sagawa et al. (2019), we do not assume access to validation and test sets with independence between the background and the label. So, we split their dataset differently to create our own training and test datasets with substantially different nuisance-label relationships. The training data has $90\%$ waterbirds on backgrounds with water and $90\%$ landbirds on backgrounds with land. The test data has this relationship flipped. We use the original image size of $224 \times 224 \times 3$. The training and test sets consist of $3510$ and $400$ samples respectively. We ensure that $p(\mathbf{y} = 1) = 0.5$ in training and test data. Thus, predicting the most frequent class achieves an accuracy of $0.5$. Cropping out the whole background requires manual segmentation. Instead, we use the pixels outside the central patch of $196 \times 196$ pixels as a nuisance in NuRD. This is a high-dimensional nuisance which impacts many existing methods negatively; see section 4. The covariates are the whole image; see fig. 1.

| Method | $p_{tr}$ | $\hat{p}_{\perp}$ | $p_{te}$ |
|---|---|---|---|
| ERM | $91 \pm 0\%$ | $-$ | $66 \pm 2\%$ |
| reweighting-NuRD | $85 \pm 1\%$ | $81 \pm 1\%$ | $83 \pm 2\%$ |

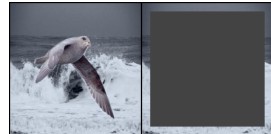

**Figure 1:** Table of results and figure showing an example of the nuisance for Waterbirds. On the left are the accuracies of NuRD and ERM on Waterbirds. Gold standard accuracy is $90\%$ (see the results paragraph below). The figure shows an image and the corresponding border nuisance. NuRD has both images during training and only the left image at test time.

**Results.**  Figure 1 reports results. We construct a gold standard model on data where waterbirds and landbirds have equal chances of appearing on backgrounds with water and land; this model achieves a test accuracy of $90\%$. ERM uses the background to predict the label, as evidenced by its test accuracy of $66\%$. Reweighting-NuRD uses the background patches to adjust for the spurious correlation to achieve an average accuracy close to the gold standard, $83\%$. We do not report generative-NuRD's performance as training on the generated images resulted in classifiers that predict as poorly as chance on real images. This may be due to the small training dataset.

## 5.4  LEARNING TO LABEL PNEUMONIA FROM X-RAYS

In many problems such as classifying cows versus penguins in natural images, the background, which is a nuisance, predicts the label. Medical imaging datasets have a similar property, where factors like the device used to take the measurement are predictive of the label but also leave a signature on the whole image. Here, we construct a dataset by mixing two chest x-ray datasets, CheXpert and MIMIC, that have different factors that affect the whole image, with or without pneumonia. The training data has $90\%$ pneumonia images from MIMIC and $90\%$ healthy images from CheXpert. The test data has the flipped relationship, with $90\%$ of the pneumonia images from CheXpert and $90\%$ of the healthy images from MIMIC. We resize the X-ray images to $32 \times 32$. Healthy cases are downsampled to make sure that in the training and test sets, healthy and pneumonia cases are equally probable. Thus, predicting the most frequent class achieves an accuracy of $0.5$. The training and test datasets consist of $12446$ and $400$ samples respectively. In chest X-rays, image segmentation cannot remove all the nuisances because nuisances like scanners alter the entire image (Pooch et al., 2019; Zech et al., 2018; Badgeley et al., 2019). However, non-lung patches, i.e. pixels outside the central patches which contain the lungs, are *a nuisance* because they do not contain physiological signals of pneumonia. We use the non-lung patches (4-pixel border) as a nuisance in NuRD. This is a high-dimensional nuisance which impacts existing methods negatively; see section 4. The covariates are the whole image; see fig. 2.

**Results.**  Figure 2 reports results. Building an oracle model in this experiment requires knowledge of all factors that correlate the label with all the parts of the X-ray. Such factors also exist within each hospital but are not recorded in MIMIC and CheXpert; for example, different departments in the same hospital can have different scanners which correlate the non-lung patches of the X-ray

| Method | $p_{tr}$ | $\hat{p}_{\perp\!\!\!\perp}$ | $p_{te}$ | | |
|---|---|---|---|---|---|
| ERM | $89 \pm 0\%$ | — | $37 \pm 1\%$ | 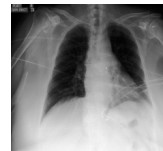 | 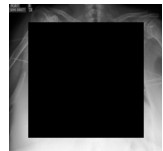 |
| generative-NURD | $70 \pm 3\%$ | $90 \pm 2$ | $41 \pm 2\%$ | | |
| reweighting-NURD | $75 \pm 1\%$ | $68 \pm 1\%$ | $61 \pm 1\%$ | | |

**Figure 2:** Table of results and figure showing an example of the nuisance for chest X-rays. The figure shows an example of a chest X-ray and the corresponding non-lung patches (right). NURD has both images during training and only the left image at test time.

with the label (Zech et al., 2018). ERM uses the nuisance to predict pneumonia, as evidenced by its test accuracy of $37\%$. Reweighting-NURD uses the non-lung patches to adjust for the spurious correlation and achieves an accuracy of $61\%$, a large improvement over ERM.

Generative-NURD also outperforms ERM's performance on average. Unlike reweighting-NURD which outperforms predicting without covariates, generative-NURD performs similar to predicting without covariates on average. The few poor test accuracies may be due to two ways generative nuisance-randomization can be imperfect: 1) little reliance of $\mathbf{x}$ on $\mathbf{z}$ with $\mathbf{y}$ fixed, 2) insufficient quality of generation which leads to poor generalization from generated to real images.

## 6 DISCUSSION

We develop an algorithm for OOD generalization in the presence of spurious correlations induced by a nuisance variable. We formalize nuisance-induced spurious correlations in a nuisance-varying family, where changing nuisance-label relationships make predictive models built from samples of one member unsuitable for other members. To identify conditional distributions that have performance guarantees on all members of the nuisance-varying family, we introduce the nuisance-randomized distribution and uncorrelating representations. We characterize one uncorrelating representation that is *simultaneously* optimal for all members. Then, we show that uncorrelating representations most informative of the label under the nuisance-randomized distribution also achieve the same optimal performance. Following this result, we propose to estimate the nuisance-randomized distribution and, under this distribution, construct the uncorrelating representation that is most informative of the label. We develop an algorithm called NURD and show that it outperforms ERM on synthetic and real data by adjusting for nuisance-induced spurious correlations. Our experiments show that NURD can use easy-to-acquire nuisances (like the border of an image) to do this adjustment; therefore, our work suggests that the need for expensive manual segmentation, even if it does help exclude all the nuisances, could be mitigated.

**Limitations and the future.** Given groups based on pairs of nuisance-label values, Sagawa et al. (2020) suggest that subsampling equally from each group produces models more robust to spurious correlations than reweighting (Byrd and Lipton, 2019; Sagawa et al., 2019); however, subsampling is ineffective when the nuisance is high-dimensional. Instead, as sufficient statistics of the conditional $p_{tr}(\mathbf{y} \mid \mathbf{z})$ render $\mathbf{y}, \mathbf{z}$ independent, grouping based on values of sufficient statistics could be promising. The nuisance-randomization steps in generative-NURD and reweighting-NURD model different distributions in the training distribution: $p_{tr}(\mathbf{x} \mid \mathbf{y}, \mathbf{z})$ and $p_{tr}(\mathbf{y} \mid \mathbf{z})$ respectively. Methods that combine the two approaches to produce better estimates of the nuisance-randomized distribution would be interesting. The first step in reweighting-NURD is to estimate $p_{tr}(\mathbf{y} \mid \mathbf{z})$. As deep networks tend to produce inflated probabilities (Guo et al., 2017), one must take care to build calibrated models for $p(\mathbf{y} \mid \mathbf{z})$. Adapting either calibration-focused losses (Kumar et al., 2018; Goldstein et al., 2020) or ensembling (Lakshminarayanan et al., 2016) may produce calibrated probabilities. In our experiments, the training data contains a single environment. Methods for invariant representation learning (Arjovsky et al., 2019; Krueger et al., 2020; Mahajan et al., 2020; Guo et al., 2021) typically require data from multiple different environments. Nuisance-randomized data has a different nuisance-label relationship from the training data, meaning it is a different environment from the training data. Following this insight, using nuisance-randomization to produce samples from different environments using data from only a single environment would a fruitful direction. The absolute performance for both ERM which exploits spurious correlations and NURD which does not, is too low to be of use in the clinic. Absolute performance could be improved with larger models, more data, using pretrained models, and multi-task learning over multiple lung conditions, all techniques that could be incorporated into learning procedures in general, including NURD.

ACKNOWLEDGEMENTS

The authors were partly supported by NIH/NHLBI Award R01HL148248, and by NSF Award 1922658 NRT-HDR: FUTURE Foundations, Translation, and Responsibility for Data Science. The authors would like to thank Mark Goldstein for helpful comments.

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

# A    FURTHER DETAILS ABOUT NuRD AND PROOFS

## A.1    DETAILS ABOUT NuRD

The algorithm boxes for reweighting-NuRD and generative-NuRD are given in algorithms 1 and 2.

**Estimating and using the weights in reweighting-NuRD.** In learning $p_{tr}(\mathbf{y} \mid \mathbf{z})$ for a high-dimensional $\mathbf{z}$, flexible models like deep neural networks can have zero training loss when the model memorizes the training data. For a discrete $\mathbf{y}$, such a model would output $\hat{p}_{tr}(\mathbf{y} \mid \mathbf{z}) = 1$ for every sample in the training data. Then, the model's weight estimates on the training data are $p(\mathbf{y})/\hat{p}_{tr}(\mathbf{y} \mid \mathbf{z}) = p(\mathbf{y})$. Weighting the training data with such estimates fails to break the nuisance-label relationship because $p_{tr}(\mathbf{y}, \mathbf{z}, \mathbf{x}) \frac{p(\mathbf{y})}{\hat{p}_{tr}(\mathbf{y} \mid \mathbf{z})} = p_{tr}(\mathbf{y}, \mathbf{z}, \mathbf{x}) p(\mathbf{y}) \propto p_{tr}(\mathbf{z} \mid \mathbf{y}) p(\mathbf{x} \mid \mathbf{y}, \mathbf{z})$. To avoid such poor weight estimation, we employ a cross-fitting procedure: split the data into $K$ disjoint folds, and the weights for each fold are produced by a model trained and validated on the rest of the folds. See algorithm 1 for details. In estimating loss for each batch under $\hat{p}_{\perp}$ during training, one can either weight the per-sample loss or produce the batches themselves via weighted sampling from the data with replacement.

**Density-ratio trick in Distillation.** The density-ratio trick for estimating mutual information (Sugiyama et al., 2012) involves Monte Carlo estimating the mutual information using a binary classifier. Let $\ell = 1$ be the pseudolabel for samples from $p_{\perp}(\mathbf{y}, r_{\gamma}(\mathbf{x}), \mathbf{z})$, and $\ell = 0$ for samples from $p_{\perp}(\mathbf{y}, r_{\gamma}(\mathbf{x})) p_{\perp}(\mathbf{z})$. Then,

$$\mathbf{I}_{\hat{p}_{\perp}}([r_{\gamma}(\mathbf{x}), \mathbf{y}]; \mathbf{z}) = \mathbb{E}_{\hat{p}_{\perp}(\mathbf{y}, \mathbf{z}, \mathbf{x})} \log \frac{\hat{p}_{\perp}(\mathbf{y}, r_{\gamma}(\mathbf{x}), \mathbf{z})}{\hat{p}_{\perp}(\mathbf{y}, r_{\gamma}(\mathbf{x})) \hat{p}_{\perp}(\mathbf{z})} = \mathbb{E}_{\hat{p}_{\perp}(\mathbf{y}, \mathbf{z}, \mathbf{x})} \log \frac{p(\ell = 1 \mid \mathbf{y}, \mathbf{z}, r_{\gamma}(\mathbf{x}))}{1 - p(\ell = 1 \mid \mathbf{y}, \mathbf{z}, r_{\gamma}(\mathbf{x}))}.$$

With parameters $\phi$, we estimate the conditional probability with a *critic model*, denoted $p_{\phi}$.

**Accounting for shifts in the marginal label distribution.** NuRD relies on the assumption in eq. (1) that distributions in the nuisance-varying family $\mathcal{F}$ have the same marginal $p(\mathbf{y})$. What happens if $p_{te}$ comes from a nuisance-varying family with a different marginal? Formally, with $p_{tr} \in \mathcal{F}$, let $p_{te}$ belong to a nuisance-varying family $\mathcal{F}' = \{p_{te}(\mathbf{y})/p_{tr}(\mathbf{y}) p_D(\mathbf{y}, \mathbf{z}, \mathbf{x}) = p_{te}(\mathbf{y}) p_D(\mathbf{z} \mid \mathbf{y}) p(\mathbf{x} \mid \mathbf{y}, \mathbf{z})\}$ where $p_D \in \mathcal{F}$. Given knowledge of the marginal distribution $p_{te}(\mathbf{y})$, note that the weighted training distribution $p_{tr}' = p_{te}(\mathbf{y})/p_{tr}(\mathbf{y}) p_{tr}(\mathbf{y}, \mathbf{z}, \mathbf{x})$ lives in $\mathcal{F}'$. Running NuRD on $p_{tr}'$ produces predictive models that generalize to $p_{te}$. To see this, note $p_{\perp}'(\mathbf{y}, \mathbf{z}, \mathbf{x}) = p_{tr}'(\mathbf{y}) p_{tr}'(\mathbf{z}) p(\mathbf{x} \mid \mathbf{y}, \mathbf{z})$ is a nuisance-randomized distribution in $\mathcal{F}'$. With $\mathcal{R}(p_{\perp}')$ as the uncorrelating set of representations defined with respect to $p_{\perp}'$, i.e. $r'(\mathbf{x}) \in \mathcal{R}(p_{\perp}') \implies \mathbf{y} \perp\!\!\!\perp_{p_{\perp}'} \mathbf{z} \mid r'$, lemma 1 and theorem 1 hold. It follows that running NuRD on samples from $p_{te}(\mathbf{y})/p_{tr}(\mathbf{y}) p_{tr}(\mathbf{y}, \mathbf{z}, \mathbf{x})$ produces an estimate of $p_{\perp}'(\mathbf{y} \mid r'(\mathbf{x}))$ $(r'(\mathbf{x}) \in \mathcal{R}(p_{\perp}'))$ with the maximal performance on every $p_{te} \in \mathcal{F}'$ if a maximally blocking $r^*(\mathbf{x}) \in \mathcal{R}(p_{\perp}')$.

## A.2    EXTENDED RELATED WORK

Domain generalization methods aim to build models with the goal of generalizing to unseen test data different from the training data (Gulrajani and Lopez-Paz, 2020). Recent work uses multiple *sufficiently different* environments to generalize to unseen test data that lies in the support of the given environments or subgroups (Li et al., 2018a; Arjovsky et al., 2019; Akuzawa et al., 2019; Mahajan et al., 2020; Krueger et al., 2020; Bellot and van der Schaar, 2020; Li et al., 2018b; Goel et al., 2020; Wald et al., 2021). Chang et al. (2020) develop a multi-environment objective to interpret neural network predictions that are robust to spurious correlations. Similarly, domain-invariant learning and related methods build representations that are independent of the domain (Ganin et al., 2016; Xie et al., 2017; Zhang et al., 2018; Ghimire et al., 2020; Adeli et al., 2021; Zhou et al., 2020).

Due to its focus on nuisances, NuRD works with data from a single environment. As in section 4, to split the data into multiple environments, one can split the data into groups based on the value of the nuisance. Then, domain-invariant methods build representations that are independent of the nuisance and under nuisance-induced spurious correlations these representations may ignore semantic features because they are correlated with the nuisance. Domain adaptation (Daumé III, 2009; Ben-David et al., 2007; Farahani et al., 2021) methods assume access to unlabelled test data which NuRD

does not require. We do not assume access to the test data because nuisance-label relationships can change over time or geography which, in turn, changes the the target distribution.

---

**Algorithm 1:** Reweighting-NURD

---

**Input:** Training data $D$, specification of the weight model $p_\alpha(\mathbf{y} \mid \mathbf{z})$ which estimates $p_{tr}(\mathbf{y} \mid \mathbf{z})$, representation model $r_\gamma(\mathbf{x})$, predictive model $p_\theta(\mathbf{y} \mid r_\gamma(\mathbf{x}))$ and critic model $p_\phi(\ell \mid \mathbf{y}, \mathbf{z}, r_\gamma(\mathbf{x}))$; regularization coefficient $\lambda$, number of iterations for the weight model $N_w$, for the predictive model and representation $N_p$, and the number of critic model steps $N_c$. Number of folds $K$.

**Result:** Return estimate of $p_\perp(\mathbf{y} \mid r_\gamma(\mathbf{x}))$ for $r_\gamma \in \mathcal{R}(p_\perp)$ with maximal information with $\mathbf{y}$.

**Nuisance Randomization step**;

Estimate the marginal distribution over the label $\hat{p}(\mathbf{y})$;

Split data into $K$ equal disjoint folds, $D = \{F_i\}_{i \leq K}$, for cross-fitting;

**for** *each fold $F_i$, $i \leq K$* **do** // (cross-fitting)

    Initialize $p_\alpha(\mathbf{y} \mid \mathbf{z})$;

    **for** $N_w$ *iterations* **do**

        Sample training batch from the rest of the folds $(F_{-i})$ : $B \sim F_{-i}$;

        Compute likelihood $\sum_{(\mathbf{y}_i, \mathbf{z}_i) \in B} \log p_\alpha(\mathbf{y}_i \mid \mathbf{z}_i)$;

        Update $\alpha$ to maximize this likelihood (via Adam for example);

    **end**

    Produce weights $w_i = \hat{p}(\mathbf{y}_i)/p_\alpha(\mathbf{y}_i \mid \mathbf{z}_i)$ for each $(\mathbf{y}_i, \mathbf{z}_i, \mathbf{x}_i) \in F_i$;

**end**

**Distillation step**;

Initialize $r_\gamma, p_\theta, p_\phi$;

**for** $N_p$ *iterations* **do**

    **for** $N_c$ *iterations* **do**

        Sample training batch $B \sim D$ and sample independent copies of $\mathbf{z}$ marginally: $\tilde{\mathbf{z}}_i \sim D$;

        Construct batch $\tilde{B} = \{\mathbf{y}_i, \tilde{\mathbf{z}}_i, \mathbf{x}_i\}$;

        Compute likelihood

$$\sum_{(\mathbf{y}_i, \mathbf{z}_i, \mathbf{x}_i) \in B} w_i \log p_\phi(\ell = 1 \mid \mathbf{y}_i, \mathbf{z}_i, r_\gamma(\mathbf{x}_i)) + \sum_{(\mathbf{y}_i, \tilde{\mathbf{z}}_i, \mathbf{x}_i) \in \tilde{B}} w_i \log p_\phi(\ell = 0 \mid \mathbf{y}_i, \tilde{\mathbf{z}}_i, r_\gamma(\mathbf{x}_i)).$$

        Update $\phi$ to maximize likelihood (via Adam for example);

    **end**

    Sample training batch $B \sim D$;

    Compute distillation objective using the density-ratio trick

$$\frac{1}{|B|} \sum_{(\mathbf{x}_i, \mathbf{y}_i, \mathbf{z}_i) \in B} w_i \left[ \log p_\theta(\mathbf{y}_i \mid r_\gamma(\mathbf{x}_i)) - \lambda \log \frac{p_\phi(\ell = 1 \mid \mathbf{y}_i, \mathbf{z}_i, r_\gamma(\mathbf{x}_i))}{1 - p_\phi(\ell = 1 \mid \mathbf{y}_i, \mathbf{z}_i, r_\gamma(\mathbf{x}_i))} \right].$$

    Update $\theta, \gamma$ to maximize objective (via Adam for example).

**end**

Return $p_\theta(\mathbf{y} \mid r_\gamma(\mathbf{x}))$.

---

Taking a distributional robustness (Duchi et al., 2021) approach, Sagawa et al. (2019) applied group-DRO to training data where the relative size of certain groups in the training data results in spurious correlations. Given these groups, group-DRO optimizes the worst error across distributions formed by weighted combinations of the groups. With high dimensional $\mathbf{z}$ as in our experiments, defining groups based on the value of the nuisance typically results in groups with at most one sample; with such groups, group-DRO will encourage memorizing the training data. Other work aims to minimize worst subgroup error with a finite number of fixed but unknown subgroups (Lahoti et al., 2020; Martinez et al., 2021); as subgroups are unknown, they only find an approximate minimizer of the worst subgroup error in general even with infinite data.

In contrast, NURD builds predictive models with performance guarantees across all test distributions (that factorize as eq. (1)) using knowledge of the nuisance. Given the nuisance, existence of a finite number of subgroups maps to an additional discreteness assumption on the nuisance variable; NURD works with general high-dimensional nuisances. Wang and Culotta (2020b) focus on

sentiment analysis of reviews and build a dataset where the nuisance label relationship is destroyed by swapping words known to be associated with sentiment of the review, with their antonyms. This is equivalent to using domain-specific knowledge to sample from $p(\mathbf{x} \mid \mathbf{y}, \mathbf{z})$ in generative NURD. NURD requires no domain-specific knowledge about the generative model $p(\mathbf{x} \mid \mathbf{y}, \mathbf{z})$.

---

**Algorithm 2:** Generative-NURD

---

**Input:** Training data $D$, specification of the generative model $p_\beta(\mathbf{x} \mid \mathbf{y}, \mathbf{z})$ that estimates $p_{tr}(\mathbf{x} \mid \mathbf{y}, \mathbf{z})$, representation model $r_\gamma(\mathbf{x})$, predictive model $p_\theta(\mathbf{y} \mid r_\gamma(\mathbf{x}))$, and critic model $p_\phi(\ell \mid \mathbf{y}, \mathbf{z}, r_\gamma(\mathbf{x}))$; regularization coefficient $\lambda$, number of iterations for the weight model $N_w$, number of iterations for the predictive model and representation $N_p$, number of critic steps $N_c$.

**Result:** Return estimate of $p_\perp(\mathbf{y} \mid r_\gamma(\mathbf{x}))$ for $r_\gamma \in \mathcal{R}(p_\perp)$ with maximal information with $\mathbf{y}$.

**Nuisance Randomization step**;

**for** $N_w$ *iterations* **do**

    Sample training batch $B \sim D$;

    Compute likelihood $\sum_{(\mathbf{y}_i, \mathbf{z}_i, \mathbf{x}_i) \in B} \log p_\beta(\mathbf{x}_i \mid \mathbf{z}_i, \mathbf{y}_i)$ (or some generative objective);

    Update $\beta$ to maximize objective above;

**end**

Estimate the marginal distribution over the label $\hat{p}(\mathbf{y})$;

Sample independent label and nuisance $\mathbf{y}_i \sim D, \mathbf{z}_j \sim D$, and then sample $\tilde{\mathbf{x}} \sim p_\beta(\mathbf{x} \mid \mathbf{y}_i, \mathbf{z}_j)$;

Construct dataset $\hat{D}$ using triples $\{\mathbf{y}_k = \mathbf{y}_i, \mathbf{z}_k = \mathbf{z}_j, \mathbf{x}_k = \hat{\mathbf{x}}\}$;

**Distillation step**;

Initialize $r_\gamma, p_\theta, p_\phi$;

**for** $N_p$ *iterations* **do**

    **for** $N_c$ *iterations* **do**

        Sample training batch $B \sim D$ and sample independent copies of $\mathbf{z}$ marginally: $\tilde{\mathbf{z}}_i \sim D$;

        Construct batch $\tilde{B} = \{\mathbf{y}_i, \tilde{\mathbf{z}}_i, \mathbf{x}_i\}$ using $B$;

        Compute likelihood

$$\sum_{(\mathbf{y}_i, \mathbf{z}_i, \mathbf{x}_i) \in B} w_i \log p_\phi(\ell = 1 \mid \mathbf{y}_i, \mathbf{z}_i, r_\gamma(\mathbf{x}_i)) + \sum_{(\mathbf{y}_i, \tilde{\mathbf{z}_i}, \mathbf{x}_i) \in \tilde{B}} w_i \log p_\phi(\ell = 0 \mid \mathbf{y}_i, \tilde{\mathbf{z}}_i, r_\gamma(\mathbf{x}_i)).$$

        Update $\phi$ to maximize likelihood (via Adam for example);

    **end**

    Sample batch from generated training data $B \sim \tilde{D}$;

    Compute distillation objective using the density-ratio trick

$$\frac{1}{|B|} \sum_{(\mathbf{x}_k, \mathbf{y}_k, \mathbf{z}_k) \in B} \left[ \log p_\theta(\mathbf{y}_k \mid r_\gamma(\mathbf{x}_k)) - \lambda \log \frac{p_\phi(\ell = 1 \mid \mathbf{y}_k, \mathbf{z}_k, r_\gamma(\mathbf{x}_k))}{1 - p_\phi(\ell = 1 \mid \mathbf{y}_k, \mathbf{z}_k, r_\gamma(\mathbf{x}_k))} \right];$$

    Update $\theta, \gamma$ to maximize objective (via Adam for example).

**end**

Return $p_\theta(\mathbf{y} \mid r_\gamma(\mathbf{x}))$.

---

## A.3 KEY LEMMAS FOR UNCORRELATING REPRESENTATIONS $r \in \mathcal{R}(p_\perp)$

In this lemma, we derive the performance of the nuisance-randomized conditional $p_\perp(\mathbf{y} \mid r(\mathbf{x}))$ for any $r \in \mathcal{R}(p_\perp)$ and show that it is at least as good as predicting without covariates on any $p_{te} \in \mathcal{F}$.

**Lemma 1.** *Let $\mathcal{F}$ be a nuisance-varying family (eq. (1)) and $p_\perp = p(\mathbf{y})p_{tr}(\mathbf{z})p(\mathbf{x} \mid \mathbf{y}, \mathbf{z})$ for some $p_{tr} \in \mathcal{F}$. Assume $\forall p_D \in \mathcal{F}$, $p_D(\mathbf{z} \mid \mathbf{y})$ is bounded. If $r(\mathbf{x}) \in \mathcal{R}(p_\perp)$, then $\forall p_{te} \in \mathcal{F}$, the performance of $p_\perp(\mathbf{y} \mid r(\mathbf{x}))$ is*

$$\texttt{Perf}_{p_{te}}(p_\perp(\mathbf{y} \mid r(\mathbf{x}))) = \texttt{Perf}_{p_{te}}(p(\mathbf{y})) + \mathop{\mathbb{E}}_{p_{te}(\mathbf{y},\mathbf{z})} KL\left[p(r(\mathbf{x}) \mid \mathbf{y}, \mathbf{z}) \,\|\, \mathbb{E}_{p(\mathbf{y})}p(r(\mathbf{x}) \mid \mathbf{y}, \mathbf{z})\right]. \quad (8)$$

*As the KL-divergence is non-negative, $\texttt{Perf}_{p_{te}}(p_\perp(\mathbf{y} \mid r(\mathbf{x}))) \geq \texttt{Perf}_{p_{te}}(p(\mathbf{y}))$.*

*Proof.* (of lemma 1) Note that the identity $\mathbb{E}_{p(\mathbf{x})}g \circ f(\mathbf{x}) = \mathbb{E}_{p(f(\mathbf{x}))}g \circ f(\mathbf{x})$ implies that

$$\mathbb{E}_{p_{te}(\mathbf{y},\mathbf{x})} \log \frac{p_{te}(\mathbf{y})}{p_\perp(\mathbf{y} \mid r(\mathbf{x}))} = \mathbb{E}_{p_{te}(\mathbf{y},r(\mathbf{x}))} \log \frac{p_{te}(\mathbf{y})}{p_\perp(\mathbf{y} \mid r(\mathbf{x}))}.$$

As $p_\perp(\mathbf{z} \mid \mathbf{y}) = p_{tr}(\mathbf{z}) > 0$ on $\mathbf{z} \in S_\mathcal{F}$ and $\mathbf{y}$ s.t. $p(\mathbf{y}) > 0$ is bounded, lemma 3 implies that $p_\perp(\mathbf{y}, \mathbf{z}, \mathbf{x}) > 0 \Leftrightarrow p_{te}(\mathbf{y}, \mathbf{z}, \mathbf{x}) > 0$. This fact implies the following **KL** terms and expectations of log-ratios are all well-defined:

$$-\texttt{Perf}_{p_{te}}(p_\perp(\mathbf{y} \mid r(\mathbf{x}))) = \mathbb{E}_{p_{te}(\mathbf{x})}\text{KL}\left[p_{te}(\mathbf{y} \mid \mathbf{x}) \,\|\, p_\perp(\mathbf{y} \mid r(\mathbf{x}))\right]$$

$$= \mathbb{E}_{p_{te}(\mathbf{y},\mathbf{x})} \log \frac{p_{te}(\mathbf{y} \mid \mathbf{x})p_{te}(\mathbf{y})}{p_\perp(\mathbf{y} \mid r(\mathbf{x}))p_{te}(\mathbf{y})}$$

$$= \mathbb{E}_{p_{te}(\mathbf{y},\mathbf{x})} \log \frac{p_{te}(\mathbf{y} \mid \mathbf{x})}{p_{te}(\mathbf{y})} + \mathbb{E}_{p_{te}(\mathbf{y},\mathbf{x})} \log \frac{p_{te}(\mathbf{y})}{p_\perp(\mathbf{y} \mid r(\mathbf{x}))}$$

$$= \mathbb{E}_{p_{te}(\mathbf{y},\mathbf{x})} \log \frac{p_{te}(\mathbf{y} \mid \mathbf{x})}{p_{te}(\mathbf{y})} + \mathbb{E}_{p_{te}(\mathbf{y},r(\mathbf{x}))} \log \frac{p(\mathbf{y})}{p_\perp(\mathbf{y} \mid r(\mathbf{x}))}$$

$$= \mathbb{E}_{p_{te}(\mathbf{x})}\text{KL}\left[p_{te}(\mathbf{y} \mid \mathbf{x}) \,\|\, p_\perp(\mathbf{y})\right] + \mathbb{E}_{p_{te}(\mathbf{y},\mathbf{z},r(\mathbf{x}))} \log \frac{p(\mathbf{y})}{p_\perp(\mathbf{y} \mid r(\mathbf{x}))}$$

$$= \mathbb{E}_{p_{te}(\mathbf{x})}\text{KL}\left[p_{te}(\mathbf{y} \mid \mathbf{x}) \,\|\, p_\perp(\mathbf{y})\right] + \mathbb{E}_{p_{te}(\mathbf{y},\mathbf{z})}\mathbb{E}_{p_\perp(r(\mathbf{x}) \mid \mathbf{y},\mathbf{z})} \log \frac{p_\perp(\mathbf{y})}{p_\perp(\mathbf{y} \mid r(\mathbf{x}))}$$

$$= \mathbb{E}_{p_{te}(\mathbf{x})}\text{KL}\left[p_{te}(\mathbf{y} \mid \mathbf{x}) \,\|\, p_\perp(\mathbf{y})\right] + \mathbb{E}_{p_{te}(\mathbf{y},\mathbf{z})}\mathbb{E}_{p_\perp(r(\mathbf{x}) \mid \mathbf{y},\mathbf{z})} \log \frac{p_\perp(\mathbf{y} \mid \mathbf{z})}{p_\perp(\mathbf{y} \mid r(\mathbf{x}),\mathbf{z})}$$

$$= \mathbb{E}_{p_{te}(\mathbf{x})}\text{KL}\left[p_{te}(\mathbf{y} \mid \mathbf{x}) \,\|\, p_\perp(\mathbf{y})\right] + \mathbb{E}_{p_{te}(\mathbf{y},\mathbf{z})}\mathbb{E}_{p_\perp(r(\mathbf{x}) \mid \mathbf{y},\mathbf{z})} \log \frac{p_\perp(\mathbf{y} \mid \mathbf{z})p_\perp(r(\mathbf{x}) \mid \mathbf{z})}{p_\perp(\mathbf{y},r(\mathbf{x}) \mid \mathbf{z})}$$

$$= \mathbb{E}_{p_{te}(\mathbf{x})}\text{KL}\left[p_{te}(\mathbf{y} \mid \mathbf{x}) \,\|\, p_\perp(\mathbf{y})\right] + \mathbb{E}_{p_{te}(\mathbf{y},\mathbf{z})}\mathbb{E}_{p_\perp(\mathbf{x} \mid \mathbf{y},\mathbf{z})} \log \frac{p_\perp(r(\mathbf{x}) \mid \mathbf{z})}{p_\perp(r(\mathbf{x}) \mid \mathbf{y},\mathbf{z})}$$

$$= \mathbb{E}_{p_{te}(\mathbf{x})}\text{KL}\left[p_{te}(\mathbf{y} \mid \mathbf{x}) \,\|\, p_\perp(\mathbf{y})\right] - \mathbb{E}_{p_{te}(\mathbf{y},\mathbf{z})}\mathbb{E}_{p_\perp(r(\mathbf{x}) \mid \mathbf{y},\mathbf{z})} \log \frac{p_\perp(r(\mathbf{x}) \mid \mathbf{y},\mathbf{z})}{p_\perp(r(\mathbf{x}) \mid \mathbf{z})}$$

$$= \mathbb{E}_{p_{te}(\mathbf{x})}\text{KL}\left[p_{te}(\mathbf{y} \mid \mathbf{x}) \,\|\, p_\perp(\mathbf{y})\right] - \mathbb{E}_{p_{te}(\mathbf{y},\mathbf{z})}\text{KL}\left[p_\perp(r(\mathbf{x}) \mid \mathbf{y},\mathbf{z}) \,\|\, p_\perp(r(\mathbf{x}) \mid \mathbf{z})\right]$$

Here, $p_\perp(r(\mathbf{x}) \mid \mathbf{y}, \mathbf{z}) = p(r(\mathbf{x}) \mid \mathbf{y}, \mathbf{z})$ as $p_\perp(\mathbf{x} \mid \mathbf{y}, \mathbf{z}) = p(\mathbf{x} \mid \mathbf{y}, \mathbf{z})$ by definition of the nuisance-varying family. The proof follows by noting that the gap in performance of $p_\perp(\mathbf{y} \mid r(\mathbf{x}))$ and $p(\mathbf{y})$ equals an expected **KL** term:

$$-\texttt{Perf}_{p_{te}}(p(\mathbf{y})) + \texttt{Perf}_{p_{te}}(p_\perp(\mathbf{y} \mid r(\mathbf{x}))) = \mathbb{E}_{p_{te}(\mathbf{y},\mathbf{z})}\text{KL}\left[p_\perp(r(\mathbf{x}) \mid \mathbf{y},\mathbf{z}) \,\|\, p_\perp(r(\mathbf{x}) \mid \mathbf{z})\right]$$

$$= \mathbb{E}_{p_{te}(\mathbf{y},\mathbf{z})}\text{KL}\left[p(r(\mathbf{x}) \mid \mathbf{y},\mathbf{z}) \,\|\, \mathbb{E}_{p(\mathbf{y})}p(r(\mathbf{x}) \mid \mathbf{y},\mathbf{z})\right]. \quad (9)$$

Rearranging these terms completes the proof. □

Lemma 2 shows that uncorrelating sets are the same for any nuisance-randomized distribution and that the conditional distribution of the label given an uncorrelating representations is the same for all nuisance-randomized distributions.

**Lemma 2.** *Let $\mathcal{F}$ be a nuisance-varying family with $p(\mathbf{y})$ and $p(\mathbf{x} \mid \mathbf{y}, \mathbf{z})$ and nuisance space $S_{\mathcal{F}}$. Consider distributions $p_{\perp,1}(\mathbf{y}, \mathbf{z}, \mathbf{x}) = p(\mathbf{y})p_{\perp,1}(\mathbf{z})p(\mathbf{x} \mid \mathbf{y}, \mathbf{z})$ and $p_{\perp,2}(\mathbf{y}, \mathbf{z}, \mathbf{x}) = p(\mathbf{y})p_{\perp,2}(\mathbf{z})p(\mathbf{x} \mid \mathbf{y}, \mathbf{z})$ such that $p_{\perp,1}(\mathbf{z}) > 0, p_{\perp,2}(\mathbf{z}) > 0$ for $\mathbf{z} \in S_{\mathcal{F}}$, and $p_{\perp,1}(\mathbf{y}, \mathbf{z}, \mathbf{x}) > 0 \iff p_{\perp,2}(\mathbf{y}, \mathbf{z}, \mathbf{x}) > 0$. Then, the uncorrelating sets are equal $\mathcal{R}(p_{\perp,1}) = \mathcal{R}(p_{\perp,2})$ and for any $r(\mathbf{x}) \in \mathcal{R}(p_{\perp,1})$,*

$$p_{\perp,1}(\mathbf{y} \mid r(\mathbf{x})) = p_{\perp,2}(\mathbf{y} \mid r(\mathbf{x})).$$

*Proof.* By the assumption that $p_{\perp,1}(\mathbf{y}, \mathbf{z}, \mathbf{x}) > 0 \Leftrightarrow p_{\perp,2}(\mathbf{y}, \mathbf{z}, \mathbf{x}) > 0$, there exist some $\mathbf{z}$ such that $p_{\perp,1}(\mathbf{z} \mid r(\mathbf{x})) > 0$ and $p_{\perp,2}(\mathbf{z} \mid r(\mathbf{x})) > 0$. With such $\mathbf{z}$, for any $r \in \mathcal{R}(p_{\perp,1})$,

$$
\begin{aligned}
p_{\perp,1}(\mathbf{y} \mid r(\mathbf{x})) &= p_{\perp,1}(\mathbf{y} \mid r(\mathbf{x}), \mathbf{z}) \\
&= p(\mathbf{y}) \frac{p(r(\mathbf{x}) \mid \mathbf{y}, \mathbf{z})}{p_{\perp,1}(r(\mathbf{x}) \mid \mathbf{z})} \\
&= p(\mathbf{y}) \frac{p(r(\mathbf{x}) \mid \mathbf{y}, \mathbf{z})}{\mathbb{E}_{p_{\perp,1}(\mathbf{y} \mid \mathbf{z})}[p_{\perp,1}(r(\mathbf{x}) \mid \mathbf{z}, \mathbf{y})]} \\
&= p(\mathbf{y}) \frac{p(r(\mathbf{x}) \mid \mathbf{y}, \mathbf{z})}{\mathbb{E}_{p(\mathbf{y})}p(r(\mathbf{x}) \mid \mathbf{z}, \mathbf{y})} \\
&= p(\mathbf{y}) \frac{p(r(\mathbf{x}) \mid \mathbf{y}, \mathbf{z})}{\mathbb{E}_{p_{\perp,2}(\mathbf{y} \mid \mathbf{z})}p(r(\mathbf{x}) \mid \mathbf{z}, \mathbf{y})} \\
&= p(\mathbf{y}) \frac{p(r(\mathbf{x}) \mid \mathbf{y}, \mathbf{z})}{p_{\perp,2}(r(\mathbf{x}) \mid \mathbf{z})} \\
&= p_{\perp,2}(\mathbf{y} \mid r(\mathbf{x}), \mathbf{z})
\end{aligned}
$$

Taking expectation on both sides with respect to $p_{\perp,2}(\mathbf{z} \mid r(\mathbf{x}))$,

$$\mathbb{E}_{p_{\perp,2}(\mathbf{z} \mid r(\mathbf{x}))}p_{\perp,1}(\mathbf{y} \mid r(\mathbf{x})) = \mathbb{E}_{p_{\perp,2}(\mathbf{z} \mid r(\mathbf{x}))}p_{\perp,2}(\mathbf{y} \mid r(\mathbf{x}), \mathbf{z}) = p_{\perp,2}(\mathbf{y} \mid r(\mathbf{x})). \tag{10}$$

Note that $\mathbb{E}_{p_{\perp,2}(\mathbf{z} \mid r(\mathbf{x}))}p_{\perp,1}(\mathbf{y} \mid r(\mathbf{x})) = p_{\perp,1}(\mathbf{y} \mid r(\mathbf{x}))$, which implies

$$p_{\perp,1}(\mathbf{y} \mid r(\mathbf{x})) = p_{\perp,1}(\mathbf{y} \mid r(\mathbf{x}), \mathbf{z}) = p_{\perp,2}(\mathbf{y} \mid r(\mathbf{x}), \mathbf{z}) = p_{\perp,2}(\mathbf{y} \mid r(\mathbf{x})),$$

completing one part of the proof, $p_{\perp,1}(\mathbf{y} \mid r(\mathbf{x})) = p_{\perp,2}(\mathbf{y} \mid r(\mathbf{x}))$.

Further, we showed $\mathbf{y} \perp\!\!\!\perp_{p_{\perp,1}} \mathbf{z} \mid r(\mathbf{x}) \implies \mathbf{y} \perp\!\!\!\perp_{p_{\perp,2}} \mathbf{z} \mid r(\mathbf{x})$ which means $r(\mathbf{x}) \in \mathcal{R}(p_{\perp,2})$. As the above proof holds with $p_{\perp,1}, p_{\perp,2}$ swapped with each other, $r(\mathbf{x}) \in \mathcal{R}(p_{\perp,1}) \iff r(\mathbf{x}) \in \mathcal{R}(p_{\perp,2})$. $\qquad\square$

The next lemma shows that every member of the nuisance-varying family is positive over the same set of $\mathbf{y}, \mathbf{z}, \mathbf{x}$ and is used in proposition 1 and lemma 1.

**Lemma 3.** *Let the nuisance-varying family $\mathcal{F}$ be defined with $p(\mathbf{y}), p(\mathbf{x} \mid \mathbf{y}, \mathbf{z})$ and nuisance space $S_{\mathcal{F}}$. Let distributions $p_D = p(\mathbf{y})p_D(\mathbf{z} \mid \mathbf{y})p(\mathbf{x} \mid \mathbf{z}, \mathbf{y})$ and $p'_D = p(\mathbf{y})p'_D(\mathbf{z} \mid \mathbf{y})p(\mathbf{x} \mid \mathbf{z}, \mathbf{y})$ be such that $p_D(\mathbf{z} \mid \mathbf{y}), p'_D(\mathbf{z} \mid \mathbf{y}) > 0$ for all $\mathbf{y}$ such that $p(\mathbf{y}) > 0$ and $\mathbf{z} \in S_{\mathcal{F}}$. Further assume $p_D(\mathbf{z} \mid \mathbf{y}), p'_D(\mathbf{z} \mid \mathbf{y})$ are bounded. Then, $p_D(\mathbf{y}, \mathbf{z}, \mathbf{x}) > 0 \Leftrightarrow p'_D(\mathbf{y}, \mathbf{z}, \mathbf{x}) > 0$.*

*Proof.* For any $\mathbf{z} \in S_{\mathcal{F}}$ and any $\mathbf{y}$ such that $p(\mathbf{y}) > 0$, $p_D(\mathbf{z} \mid \mathbf{y}) > 0$ and $\frac{p'_D(\mathbf{z} \mid \mathbf{y})}{p_D(\mathbf{z} \mid \mathbf{y})} > 0$,

$$p'_D(\mathbf{y}, \mathbf{z}, \mathbf{x}) = p(\mathbf{x} \mid \mathbf{y}, \mathbf{z})p'_D(\mathbf{z} \mid \mathbf{y})p(\mathbf{y}) = p(\mathbf{x} \mid \mathbf{y}, \mathbf{z})p_D(\mathbf{z} \mid \mathbf{y})p(\mathbf{y})\frac{p'_D(\mathbf{z} \mid \mathbf{y})}{p_D(\mathbf{z} \mid \mathbf{y})} = p_D(\mathbf{y}, \mathbf{z}, \mathbf{x})\frac{p'_D(\mathbf{z} \mid \mathbf{y})}{p_D(\mathbf{z} \mid \mathbf{y})}. \tag{11}$$

Thus, for all $\mathbf{z}$ in the nuisance space $S_{\mathcal{F}}$ and any $\mathbf{y}$ such that $p(\mathbf{y}) > 0$,

$$p'_D(\mathbf{y}, \mathbf{z}, \mathbf{x}) > 0 \iff p_D(\mathbf{y}, \mathbf{z}, \mathbf{x}) > 0.$$

As $\mathbf{z}$ only takes values in the nuisance space $S_{\mathcal{F}}$, when $p(\mathbf{y}) = 0$,

$$p'_D(\mathbf{y}, \mathbf{z}, \mathbf{x}) = p_D(\mathbf{y}, \mathbf{z}, \mathbf{x}) = 0.$$

Together, the two statements above imply

$$p_D(\mathbf{y}, \mathbf{z}, \mathbf{x}) > 0 \Leftrightarrow p'_D(\mathbf{y}, \mathbf{z}, \mathbf{x}) > 0.$$

$\qquad\square$

### A.4 OPTIMAL UNCORRELATING REPRESENTATIONS

**Theorem 1.** *Let $r^* \in \mathcal{R}(p_\perp)$ be **maximally blocking**: $\forall r \in \mathcal{R}(p_\perp), \quad \mathbf{y} \perp\!\!\!\perp_{p_\perp} r(\mathbf{x}) \mid \mathbf{z}, r^*(\mathbf{x})$. Then,*

1. *(Simultaneous optimality)* $\forall p_{te} \in \mathcal{F}, \forall r \in \mathcal{R}(p_\perp), \quad \texttt{Perf}_{p_{te}}(r^*(\mathbf{x})) \geq \texttt{Perf}_{p_{te}}(r(\mathbf{x}))$.

2. *(Information maximality)* $\forall r(\mathbf{x}) \in \mathcal{R}(p_\perp), \quad \mathbf{I}_{p_\perp}(\mathbf{y}; r^*(\mathbf{x})) \geq \mathbf{I}_{p_\perp}(\mathbf{y}; r(\mathbf{x}))$.

3. *(Information maximality implies simultaneous optimality)* $\forall r' \in \mathcal{R}(p_\perp)$,

$$\mathbf{I}_{p_\perp}(\mathbf{y}; r'(\mathbf{x})) = \mathbf{I}_{p_\perp}(\mathbf{y}; r^*(\mathbf{x})) \implies \forall p_{te} \in \mathcal{F}, \quad \texttt{Perf}_{p_{te}}(r^*(\mathbf{x})) = \texttt{Perf}_{p_{te}}(r'(\mathbf{x})).$$

*Proof.* (proof for theorem 1)

We first prove that for any pair $r, r_2 \in \mathcal{R}(p_\perp)$ such that $r_2$ blocks $r$, $r(\mathbf{x}) \perp\!\!\!\perp_{p_\perp} \mathbf{y} \mid \mathbf{z}, r_2(\mathbf{x})$, $r_2$ dominates the performance of $r$ on every $p_{te} \in \mathcal{F}$. The simultaneously optimality of the maximally blocking representation will follow. For readability, let $\ell(r_2) = \mathbb{E}_{p_{te}(\mathbf{x})} \text{KL}\left[p_{te}(\mathbf{y} \mid \mathbf{x}) \,\|\, p_\perp(\mathbf{y} \mid r_2(\mathbf{x}))\right]$. We will show that

$$\mathbb{E}_{p_{te}(\mathbf{x})} \text{KL}\left[p_{te}(\mathbf{y} \mid \mathbf{x}) \,\|\, p_\perp(\mathbf{y} \mid r(\mathbf{x}))\right] \geq \ell(r_2).$$

We will use the following identity which follows from the fact that $p(\mathbf{x} \mid \mathbf{y}, \mathbf{z})$ does not change between distributions in the data generating process eq. (1):

$$p_D(r_2(\mathbf{x}) \mid \mathbf{y}, \mathbf{z}, r(\mathbf{x})) = \frac{p_D(r_2(\mathbf{x}), r(\mathbf{x}) \mid \mathbf{y}, \mathbf{z})}{p_D(r(\mathbf{x}) \mid \mathbf{y}, \mathbf{z})}$$

$$= \frac{p(r_2(\mathbf{x}), r(\mathbf{x}) \mid \mathbf{y}, \mathbf{z})}{p(r(\mathbf{x}) \mid \mathbf{y}, \mathbf{z})}$$

$$= p(r_2(\mathbf{x}) \mid \mathbf{y}, \mathbf{z}, r(\mathbf{x})).$$

Next, we will show that

$$\mathbb{E}_{p_{te}(\mathbf{x})} \text{KL}\left[p_{te}(\mathbf{y} \mid \mathbf{x}) \,\|\, p_\perp(\mathbf{y} \mid r(\mathbf{x}))\right]$$
$$= \ell(r_2) + \mathbb{E}_{p_{te}(\mathbf{y}, \mathbf{z}, r(\mathbf{x}))} \text{KL}\left[p_\perp(r_2(\mathbf{x}) \mid \mathbf{y}, \mathbf{z}, r(\mathbf{x})) \,\|\, p_\perp(r_2(\mathbf{x}) \mid r(\mathbf{x}), \mathbf{z})\right].$$

The steps are similar to lemma 1's proof

$$\mathbb{E}_{p_{te}(\mathbf{x})} \text{KL}\left[p_{te}(\mathbf{y} \mid \mathbf{x}) \,\|\, p_\perp(\mathbf{y} \mid r(\mathbf{x}))\right] = \mathbb{E}_{p_{te}(\mathbf{y}, \mathbf{x})} \log \frac{p_{te}(\mathbf{y} \mid \mathbf{x})}{p_\perp(\mathbf{y} \mid r_2(\mathbf{x}))} + \mathbb{E}_{p_{te}(\mathbf{y}, \mathbf{x})} \log \frac{p_\perp(\mathbf{y} \mid r_2(\mathbf{x}))}{p_\perp(\mathbf{y} \mid r(\mathbf{x}))}$$

$$= \ell(r_2) + \mathbb{E}_{p_{te}(\mathbf{y}, r(\mathbf{x}), r_2(\mathbf{x}))} \log \frac{p_\perp(\mathbf{y} \mid r_2(\mathbf{x}))}{p_\perp(\mathbf{y} \mid r(\mathbf{x}))}$$

$$= \ell(r_2) + \mathbb{E}_{p_{te}(\mathbf{y}, \mathbf{z}, r(\mathbf{x}), r_2(\mathbf{x}))} \log \frac{p_\perp(\mathbf{y} \mid r_2(\mathbf{x}), \mathbf{z})}{p_\perp(\mathbf{y} \mid r(\mathbf{x}), \mathbf{z})} \quad \{\text{as } r, r_2 \in \mathcal{R}(p_\perp)\}$$

$$= \ell(r_2) + \mathbb{E}_{p_{te}(\mathbf{y}, \mathbf{z}, r(\mathbf{x}), r_2(\mathbf{x}))} \log \frac{p_\perp(\mathbf{y} \mid r_2(\mathbf{x}), r(\mathbf{x}), \mathbf{z})}{p_\perp(\mathbf{y} \mid r(\mathbf{x}), \mathbf{z})} \quad \{\mathbf{y} \perp\!\!\!\perp_{p_\perp} r(\mathbf{x}) \mid \mathbf{z}, r_2(\mathbf{x})\}$$

$$= \ell(r_2) + \mathbb{E}_{p_{te}(\mathbf{y}, \mathbf{z}, r(\mathbf{x}), r_2(\mathbf{x}))} \log \frac{p_\perp(\mathbf{y}, r_2(\mathbf{x}) \mid r(\mathbf{x}), \mathbf{z})}{p_\perp(\mathbf{y} \mid r(\mathbf{x}), \mathbf{z}) p_\perp(r_2(\mathbf{x}) \mid r(\mathbf{x}), \mathbf{z})}$$

$$= \ell(r_2) + \mathbb{E}_{p_{te}(\mathbf{y}, \mathbf{z}, r(\mathbf{x}), r_2(\mathbf{x}))} \log \frac{p_\perp(r_2(\mathbf{x}) \mid \mathbf{y}, r(\mathbf{x}), \mathbf{z})}{p_\perp(r_2(\mathbf{x}) \mid r(\mathbf{x}), \mathbf{z})}$$

$$= \ell(r_2) + \mathbb{E}_{p_{te}(\mathbf{y}, \mathbf{z}, r(\mathbf{x}))} \mathbb{E}_{p_{te}(r_2(\mathbf{x}) \mid \mathbf{y}, \mathbf{z}, r(\mathbf{x}))} \log \frac{p_\perp(r_2(\mathbf{x}) \mid \mathbf{y}, r(\mathbf{x}), \mathbf{z})}{p_\perp(r_2(\mathbf{x}) \mid r(\mathbf{x}), \mathbf{z})}$$

$$= \ell(r_2) + \mathbb{E}_{p_{te}(\mathbf{y}, \mathbf{z}, r(\mathbf{x}))} \text{KL}\left[p_\perp(r_2(\mathbf{x}) \mid \mathbf{y}, \mathbf{z}, r(\mathbf{x})) \,\|\, p_\perp(r_2(\mathbf{x}) \mid r(\mathbf{x}), \mathbf{z})\right]$$

Noting that **KL** is non-negative and that `Perf` is negative-**KL** proves the theorem:

$$\mathbb{E}_{p_{te}(\mathbf{x})} \text{KL}\left[p_{te}(\mathbf{y} \mid \mathbf{x}) \,\|\, p_\perp(\mathbf{y} \mid r_2(\mathbf{x}))\right] \leq \mathbb{E}_{p_{te}(\mathbf{x})} \text{KL}\left[p_{te}(\mathbf{y} \mid \mathbf{x}) \,\|\, p_\perp(\mathbf{y} \mid r(\mathbf{x}))\right]. \quad (12)$$

It follows that for a maximally blocking $r^*$

$$\forall r \in \mathcal{R}(p_\perp) \quad \mathbb{E}_{p_{te}(\mathbf{x})} \text{KL}\left[p_{te}(\mathbf{y} \mid \mathbf{x}) \,\|\, p_\perp(\mathbf{y} \mid r^*(\mathbf{x}))\right] \leq \mathbb{E}_{p_{te}(\mathbf{x})} \text{KL}\left[p_{te}(\mathbf{y} \mid \mathbf{x}) \,\|\, p_\perp(\mathbf{y} \mid r(\mathbf{x}))\right].$$

As performance is negative **KL**, the proof follows that $r^*$ dominates $r$ in performance. This concludes the first part of the proof.

For the second part, we prove information maximality of a maximally blocking $r^*(\mathbf{x}) \in \mathcal{R}(p_\perp)$. The proof above shows that the model $p_\perp(\mathbf{y} \mid r^*(\mathbf{x}))$ performs at least as well as $p_\perp(\mathbf{y} \mid r(\mathbf{x}))$ for any $r(\mathbf{x}) \in \mathcal{R}(p_\perp)$ on any $p_{te} \in \mathcal{F}$. We characterize the gap in performance between $p_\perp(\mathbf{y} \mid r^*(\mathbf{x}))$ and $p_\perp(\mathbf{y} \mid r(\mathbf{x}))$ for any $r(\mathbf{x}) \in \mathcal{R}(p_\perp)$ as the following conditional mutual information term:

$$\mathbb{E}_{p_\perp(\mathbf{y},\mathbf{z},r(\mathbf{x}))}\mathrm{KL}\left[p_\perp(r^*(\mathbf{x}) \mid \mathbf{y},\mathbf{z},r(\mathbf{x})) \parallel p_\perp(r^*(\mathbf{x}) \mid r(\mathbf{x}),\mathbf{z})\right] = \mathbf{I}_{p_\perp}(r^*(\mathbf{x}); \mathbf{y} \mid \mathbf{z}, r(\mathbf{x})).$$

The entropy decomposition of conditional mutual information (with $\mathbf{H}_q(\cdot)$ as the entropy under a distribution $q$) gives two mutual information terms.

$$\mathbb{E}_{p_\perp(\mathbf{y},\mathbf{z},r(\mathbf{x}))}\mathrm{KL}\left[p_\perp(r^*(\mathbf{x}) \mid \mathbf{y},\mathbf{z},r(\mathbf{x})) \parallel p_\perp(r^*(\mathbf{x}) \mid r(\mathbf{x}),\mathbf{z})\right] = \mathbf{I}_{p_\perp}(r^*(\mathbf{x}); \mathbf{y} \mid \mathbf{z}, r(\mathbf{x})),$$

$$= \mathbf{H}_{p_\perp}(\mathbf{y} \mid \mathbf{z}, r(\mathbf{x})) - \mathbf{H}_{p_\perp}(\mathbf{y} \mid \mathbf{z}, r(\mathbf{x}), r^*(\mathbf{x}))$$

$$= \mathbf{H}_{p_\perp}(\mathbf{y} \mid \mathbf{z}, r(\mathbf{x})) - \mathbf{H}_{p_\perp}(\mathbf{y} \mid \mathbf{z}, r^*(\mathbf{x})) \qquad \{\mathbf{y} \perp\!\!\!\perp_{p_\perp} r(\mathbf{x}) \mid \mathbf{z}, r^*(\mathbf{x})\}$$

$$= \mathbf{H}_{p_\perp}(\mathbf{y} \mid r(\mathbf{x})) - \mathbf{H}_{p_\perp}(\mathbf{y} \mid r^*(\mathbf{x})) \qquad \{r, r^* \in \mathcal{R}(p_\perp)\}$$

$$= \mathbf{H}_{p_\perp}(\mathbf{y} \mid r(\mathbf{x})) - \mathbf{H}_{p_\perp}(\mathbf{y}) + \mathbf{H}_{p_\perp}(\mathbf{y}) - \mathbf{H}_{p_\perp}(\mathbf{y} \mid r^*(\mathbf{x}))$$

$$= \mathbf{I}_{p_\perp}(\mathbf{y}; r^*(\mathbf{x})) - \mathbf{I}_{p_\perp}(\mathbf{y}, r(\mathbf{x})).$$

This difference is non-negative for any $r \in \mathcal{R}(p_\perp)$ which proves the second part of the theorem:

$$\mathbf{I}_{p_\perp}(\mathbf{y}; r^*(\mathbf{x})) - \mathbf{I}_{p_\perp}(\mathbf{y}, r(\mathbf{x})) = \mathbf{I}_{p_\perp}(r^*(\mathbf{x}); \mathbf{y} \mid \mathbf{z}, r(\mathbf{x})) \geq 0.$$

For the third part, note that any representation $r'$ which satisfies $\mathbf{I}_{p_\perp}(\mathbf{y}; r^*(\mathbf{x})) = \mathbf{I}_{p_\perp}(\mathbf{y}, r'(\mathbf{x}))$ (information-equivalence) also satisfies

$$\mathbf{I}_{p_\perp}(\mathbf{y}; r^*(\mathbf{x}) \mid \mathbf{z}, r'(\mathbf{x})) = 0 \implies \mathbf{y} \perp\!\!\!\perp_{p_\perp} r^*(\mathbf{x}) \mid \mathbf{z}, r'(\mathbf{x}).$$

Under this condition, eq. (12) implies

$$\mathbb{E}_{p_{te}(\mathbf{x})}\mathrm{KL}\left[p_{te}(\mathbf{y} \mid \mathbf{x}) \parallel p_\perp(\mathbf{y} \mid r^*(\mathbf{x}))\right] \geq \mathbb{E}_{p_{te}(\mathbf{x})}\mathrm{KL}\left[p_{te}(\mathbf{y} \mid \mathbf{x}) \parallel p_\perp(\mathbf{y} \mid r'(\mathbf{x}))\right].$$

However, as $r' \in \mathcal{R}(p_\perp)$ and that $r^*(\mathbf{x})$ is maximally blocking, which (by the proof above) implies

$$\mathbb{E}_{p_{te}(\mathbf{x})}\mathrm{KL}\left[p_{te}(\mathbf{y} \mid \mathbf{x}) \parallel p_\perp(\mathbf{y} \mid r^*(\mathbf{x}))\right] \leq \mathbb{E}_{p_{te}(\mathbf{x})}\mathrm{KL}\left[p_{te}(\mathbf{y} \mid \mathbf{x}) \parallel p_\perp(\mathbf{y} \mid r'(\mathbf{x}))\right].$$

The only way both these conditions hold is if

$$\mathbb{E}_{p_{te}(\mathbf{x})}\mathrm{KL}\left[p_{te}(\mathbf{y} \mid \mathbf{x}) \parallel p_\perp(\mathbf{y} \mid r^*(\mathbf{x}))\right] = \mathbb{E}_{p_{te}(\mathbf{x})}\mathrm{KL}\left[p_{te}(\mathbf{y} \mid \mathbf{x}) \parallel p_\perp(\mathbf{y} \mid r'(\mathbf{x}))\right].$$

This completes the proof that for any $r' \in \mathcal{R}(p_\perp)$ that is information-equivalent to $r^*(\mathbf{x})$ under $p_\perp$, the model $p_\perp(\mathbf{y} \mid r'(\mathbf{x}))$ has the same performance as $p_\perp(\mathbf{y} \mid r^*(\mathbf{x}))$ for every $p_{te} \in \mathcal{F}$, and consequently, $r'$ is also optimal.

$\square$

## A.5 Minimax optimality

**Proposition 1.** *Consider a nuisance-varying family $\mathcal{F}$ (eq. (1)) such that for some $p_{tr} \in \mathcal{F}$ there exists a distribution $p_\perp \in \mathcal{F}$ such that $p_\perp = p(\mathbf{y})p_{tr}(\mathbf{z})p(\mathbf{x} \mid \mathbf{y}, \mathbf{z}) \in \mathcal{F}$. Let $\mathcal{F}$ satisfy*

$$\mathbf{y} \not\perp\!\!\!\perp_{p_D} \mathbf{z} \implies \exists p'_D \in \mathcal{F} s.t. \left[\mathbb{E}_{p'_D(\mathbf{x})}KL\left[p'_D(\mathbf{y} \mid \mathbf{x}) \parallel p_D(\mathbf{y} \mid \mathbf{x})\right] - \mathbf{I}_{p'_D}(\mathbf{x}; \mathbf{y})\right] > 0. \tag{13}$$

*If $\mathbf{y} \perp\!\!\!\perp_{p_\perp} \mathbf{z} \mid \mathbf{x}$, then $p_\perp(\mathbf{y} \mid \mathbf{x})$ is minimax optimal :*

$$p_\perp(\mathbf{y} \mid \mathbf{x}) = \underset{p_D(\mathbf{y} \mid \mathbf{x}); p_D \in \mathcal{F}}{\arg\min} \max_{p'_D \in \mathcal{F}} \mathbb{E}_{p'_D(\mathbf{x})}KL\left[p'_D(\mathbf{y} \mid \mathbf{x}) \parallel p_D(\mathbf{y} \mid \mathbf{x})\right].$$

*Proof.* (of proposition 1) By lemma 3, as $p_D(\mathbf{y}, \mathbf{z}, \mathbf{x}) > 0 \Leftrightarrow p'_D(\mathbf{y}, \mathbf{z}, \mathbf{x}) > 0$, performance is well defined for any $p_D(\mathbf{y} \mid \mathbf{x})$ on any $p'_D \in \mathcal{F}$. First, lemma 1 with $p_{te} = p'_D$ and $r(\mathbf{x}) = \mathbf{x}$ gives

$$\mathbf{I}_{p'_D}(\mathbf{x}; \mathbf{y}) - \mathbb{E}_{p'_D(\mathbf{x})}\mathrm{KL}\left[p'_D(\mathbf{y} \mid \mathbf{x}) \parallel p_\perp(\mathbf{y} \mid \mathbf{x})\right]$$

$$= \mathbb{E}_{p'_D(\mathbf{x})}\mathrm{KL}\left[p'_D(\mathbf{y} \mid \mathbf{x}) \parallel p_\perp(\mathbf{y})\right] - \mathbb{E}_{p'_D(\mathbf{x})}\mathrm{KL}\left[p'_D(\mathbf{y} \mid \mathbf{x}) \parallel p_\perp(\mathbf{y} \mid \mathbf{x})\right]$$

$$= \mathbb{E}_{p'_D(\mathbf{y},\mathbf{z})}\mathrm{KL}\left[p(\mathbf{x} \mid \mathbf{y}, \mathbf{z}) \parallel \mathbb{E}_{p(\mathbf{y})}p(\mathbf{x} \mid \mathbf{y}, \mathbf{z})\right].$$

$$= \mathbb{E}_{p'_D(\mathbf{y},\mathbf{z})}\mathrm{KL}\left[p_\perp(\mathbf{x} \mid \mathbf{y}, \mathbf{z}) \parallel p_\perp(\mathbf{x} \mid \mathbf{z})\right] \geq 0. \tag{14}$$

Thus, unlike any $p_D \in \mathcal{F}$ such that $\mathbf{y} \not\!\perp\!\!\!\perp_{p_D} \mathbf{z}$,

$$\max_{p'_D \in \mathcal{F}} \left[ \mathbb{E}_{p'_D(\mathbf{x})} \mathrm{KL} \left[ p'_D(\mathbf{y} \mid \mathbf{x}) \parallel p_\perp(\mathbf{y} \mid \mathbf{x}) \right] - \mathbf{I}_{p'_D}(\mathbf{x}; \mathbf{y}) \right] \leq 0. \qquad (15)$$

For any $p_D$ such that $\mathbf{y} \not\!\perp\!\!\!\perp_{p_D} \mathbf{z}$, let $p'_D$ be such that $\mathbb{E}_{p'_D(\mathbf{x})} \mathrm{KL} \left[ p'_D(\mathbf{y} \mid \mathbf{x}) \parallel p_D(\mathbf{y} \mid \mathbf{x}) \right] - \mathbf{I}_{p'_D}(\mathbf{x}; \mathbf{y}) > 0$. As eq. (15) implies $\mathbb{E}_{p'_D(\mathbf{x})} \mathrm{KL} \left[ p'_D(\mathbf{y} \mid \mathbf{x}) \parallel p_\perp(\mathbf{y} \mid \mathbf{x}) \right] - \mathbf{I}_{p'_D}(\mathbf{x}; \mathbf{y}) \leq 0$, it follows that $\forall p_D$ such that $\mathbf{y} \not\!\perp\!\!\!\perp_{p_D} \mathbf{z}$,

$$\max_{p'_D \in \mathcal{F}} \mathbb{E}_{p'_D(\mathbf{x})} \mathrm{KL} \left[ p'_D(\mathbf{y} \mid \mathbf{x}) \parallel p_D(\mathbf{y} \mid \mathbf{x}) \right] > \max_{p'_D \in \mathcal{F}} \mathbb{E}_{p'_D(\mathbf{x})} \mathrm{KL} \left[ p'_D(\mathbf{y} \mid \mathbf{x}) \parallel p_\perp(\mathbf{y} \mid \mathbf{x}) \right].$$

By lemma 3, any $p_D \in \mathcal{F}$ is positive over the same set of $\mathbf{y}, \mathbf{z}, \mathbf{x}$ and if $\mathbf{y} \perp\!\!\!\perp_{p_D} \mathbf{z}$, then $p_D(\mathbf{y} \mid \mathbf{x}) = p_\perp(\mathbf{y} \mid \mathbf{x})$ (see lemma 2 for proof with instantiation $r(\mathbf{x}) = \mathbf{x}$). This means

$$p_\perp(\mathbf{y} \mid \mathbf{x}) = \arg \min_{p_D \in \mathcal{F}} \max_{p'_D \in \mathcal{F}} \mathbb{E}_{p'_D(\mathbf{x})} \mathrm{KL} \left[ p'_D(\mathbf{y} \mid \mathbf{x}) \parallel p_D(\mathbf{y} \mid \mathbf{x}) \right].$$

$\square$

See proposition 3 for an example nuisance-varying family where the information criterion in eq. (13) holds.

### A.6 Distillation details and a local optima example for eq. (5)

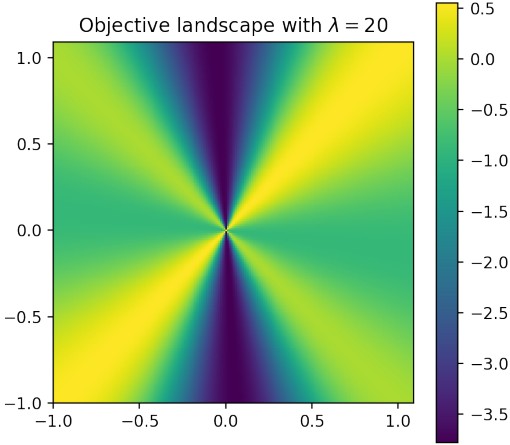

**Figure 3:** Landscape of the objective in eq. (5) for the example in eq. (2) for linear representations $r_{u,v}(\mathbf{x}) = u\mathbf{x}_1 + v\mathbf{x}_2$. Local maxima correspond to representations $r_{-u,u}$ and global maxima to representations $r_{u,u}$.

The objective in eq. (5) can have local optima when the representation is a function of the nuisance and the exogenous noise in the generation of the covariates given the nuisance and the label. Formally, let the exogenous noise $\boldsymbol{\epsilon}$ satisfy $(\boldsymbol{\epsilon}, \mathbf{z}) \perp\!\!\!\perp_{p_\perp} \mathbf{y}$. Then,

$$(\boldsymbol{\epsilon}, \mathbf{z}) \perp\!\!\!\perp_{p_\perp} \mathbf{y} \implies (f(\boldsymbol{\epsilon}, \mathbf{z}), \mathbf{z}) \perp\!\!\!\perp_{p_\perp} \mathbf{y} \implies \mathbf{z} \perp\!\!\!\perp_{p_\perp} \mathbf{y} \mid f(\mathbf{z}, \boldsymbol{\epsilon}).$$

Such a representation $r(\mathbf{x}) = f(\boldsymbol{\epsilon}, \mathbf{z})$ is both in the uncorrelating set and independent of the label $\mathbf{y}$ under the nuisance-randomized distribution $p_\perp$ meaning it does not predict the label.

**Local optima example for conditional information regularization eq. (5).** Figure 3 plots the value of the objective in eq. (5) computed analytically for $\lambda = 20$, over the class of linear representations indexed by $u, v \in \mathbf{R}$, $r_{u,v}(\mathbf{x}) = u\mathbf{x}_1 + v\mathbf{x}_2$, under the data generating process in eq. (2). Representations of the kind $r_{-u,u}(\mathbf{x}) = u(\mathbf{x}_2 - \mathbf{x}_1)$ are functions of $\mathbf{z}$ and some noise independent of the label and, as fig. 3 shows, are local maxima on the landscape of the maximization objective in eq. (5). Global maxima correspond to representations $r_{u,u}$.

**Performance characterization for jointly independent representations**

**Lemma 4.** *Let $\mathcal{F}$ be a nuisance varying family. For any jointly independent representation $r$, i.e. $[r(\mathbf{x}), \mathbf{y}] \perp\!\!\!\perp_{p_\perp} \mathbf{z}$,*

$$\forall p_{te} \in \mathcal{F} \qquad \mathtt{Perf}_{p_{te}}(p_\perp(\mathbf{y} \mid r(\mathbf{x}))) = C_{p_{te}} + \mathbf{I}_{p_\perp}(r(\mathbf{x}); \mathbf{y}),$$

*where $C_{p_{te}}$ is a $p_{te}$-dependent constant that does not vary with $r(\mathbf{x})$.*

NURD *maximizes the information term $\mathbf{I}_{p_\perp}(\mathbf{y}; r_\gamma(\mathbf{x}))$ and, therefore, maximizes performance on every member of $\mathcal{F}$ simultaneously. It follows that within the set of jointly independent representations,* NURD*, at optimality, produces a representation that is simultaneously optimal on every $p_{te} \in \mathcal{F}$.*

*Proof.* Lemma 1 says that for any uncorrelating representation $r \in \mathcal{R}(p_\perp)$ and $\forall p_{te} \in \mathcal{F}$,

$$\mathtt{Perf}_{p_{te}}(p_\perp(\mathbf{y} \mid r(\mathbf{x}))) = \mathtt{Perf}_{p_{te}}(p(\mathbf{y})) + \mathop{\mathbb{E}}_{p_{te}(\mathbf{y},\mathbf{z})} \mathrm{KL}\left[p(r(\mathbf{x}) \mid \mathbf{y}, \mathbf{z}) \,\|\, \mathbb{E}_{p(\mathbf{y})}p(r(\mathbf{x}) \mid \mathbf{y}, \mathbf{z})\right].$$

However, as the joint independence $[r(\mathbf{x}), \mathbf{y}] \perp\!\!\!\perp_{p_\perp} \mathbf{z}$ implies both the uncorrelating property and $r(\mathbf{x}) \perp\!\!\!\perp_{p_\perp} \mathbf{z} \mid \mathbf{y}$, the second term in the RHS above can be expressed as $\mathbf{I}_{p_\perp}(r(\mathbf{x}); \mathbf{y})$:

$$\mathop{\mathbb{E}}_{p_{te}(\mathbf{y},\mathbf{z})} \mathrm{KL}\left[p(r(\mathbf{x}) \mid \mathbf{y}, \mathbf{z}) \,\|\, \mathbb{E}_{p(\mathbf{y})}p(r(\mathbf{x}) \mid \mathbf{y}, \mathbf{z})\right]$$

$$= \mathop{\mathbb{E}}_{p_{te}(\mathbf{y},\mathbf{z})} \mathrm{KL}\left[p_\perp(r(\mathbf{x}) \mid \mathbf{y}, \mathbf{z}) \,\|\, \mathbb{E}_{p_\perp(\mathbf{y})}p_\perp(r(\mathbf{x}) \mid \mathbf{y}, \mathbf{z})\right]$$

$$= \mathop{\mathbb{E}}_{p_{te}(\mathbf{y},\mathbf{z})} \mathrm{KL}\left[p_\perp(r(\mathbf{x}) \mid \mathbf{y}) \,\|\, \mathbb{E}_{p_\perp(\mathbf{y})}p_\perp(r(\mathbf{x}) \mid \mathbf{y})\right]$$

$$= \mathop{\mathbb{E}}_{p_{te}(\mathbf{y},\mathbf{z})} \mathrm{KL}\left[p_\perp(r(\mathbf{x}) \mid \mathbf{y}) \,\|\, p_\perp(r(\mathbf{x}))\right]$$

$$= \mathop{\mathbb{E}}_{p_{te}(\mathbf{y})} \mathrm{KL}\left[p_\perp(r(\mathbf{x}) \mid \mathbf{y}) \,\|\, p_\perp(r(\mathbf{x}))\right]$$

$$= \mathop{\mathbb{E}}_{p_\perp(\mathbf{y})} \mathrm{KL}\left[p_\perp(r(\mathbf{x}) \mid \mathbf{y}) \,\|\, p_\perp(r(\mathbf{x}))\right]$$

$$= \mathbf{I}_{p_\perp}(r(\mathbf{x}); \mathbf{y}).$$

Noting $C_{p_{te}} = \mathtt{Perf}_{p_{te}}(p(\mathbf{y}))$ does not vary with $r(\mathbf{x})$ completes the proof. $\qquad\square$

**Performance gaps between jointly independent representations and uncorrelating representations.** The joint independence $[r(\mathbf{x}), \mathbf{y}] \perp\!\!\!\perp_{p_\perp} \mathbf{z}$ implies the uncorrelating property but uncorrelating representations only satisfy this joint independence when they are independent of the nuisance. Thus, representations that satisfy joint independence form a subset of uncorrelating representations. This begs a question: is there a loss in performance by restricting NURD to representations that satisfy said joint independence? In appendix A.7.1, we use the theory of minimal sufficient statistics (Lehmann and Casella, 2006) to show that there exists a nuisance-varying family where the best uncorrelating representation dominates every representation that satisfies joint independence on every member distribution, and is strictly better in at least one.

## A.7 COUNTERFACTUAL INVARIANCE VS. THE UNCORRELATING PROPERTY

We A) show that counterfactually invariant representations are a subset of uncorrelating representations by reducing counterfactual invariance to the joint independence $[r(\mathbf{x}), \mathbf{y}] \perp\!\!\!\perp_{p_\perp} \mathbf{z}$ and B) give an example nuisance-varying family $\mathcal{F}$ where the best uncorrelating representation strictly dominates every jointly independent representation in performance on every test distribution $p_{te} \in \mathcal{F}$: at least as good on all $p_{te} \in \mathcal{F}$ and strictly better on at least one.

We show A by proving counterfactually invariant representations satisfy joint independence $[r(\mathbf{x}), \mathbf{y}] \perp\!\!\!\perp_{p_\perp} \mathbf{z}$ which implies the uncorrelating property, but not vice versa. Counterfactual invariance implies that for all $p_D \in \mathcal{F}$, the conditional independence $r(\mathbf{x}) \perp\!\!\!\perp_{p_D} \mathbf{z} \mid \mathbf{y}$ holds by theorem 3.2 in Veitch et al. (2021). As $\mathbf{y} \perp\!\!\!\perp_{p_\perp} \mathbf{z}$, it follows that $[r(\mathbf{x}), \mathbf{y}] \perp\!\!\!\perp_{p_\perp} \mathbf{z}$; this joint independence implies the uncorrelating property, $\mathbf{y} \perp\!\!\!\perp_{p_\perp} \mathbf{z} \mid r(\mathbf{x})$. But, uncorrelating representations only satisfy the said joint independence when they are independent of the nuisance.

We show B in appendix A.7.1 by constructing a nuisance-varying family where the optimal performance is achieved by an uncorrelating representation that is *dependent* on the nuisance.

A.7.1 JOINT INDEPENDENCE VS. THE UNCORRELATING PROPERTY

Here, we discuss the performance gap between representations that are uncorrelating $(\mathbf{y} \perp\!\!\!\perp_{p_\perp} \mathbf{z} \mid r(\mathbf{x}))$ and those that satisfy the joint independence $(\mathbf{y}, r(\mathbf{x})) \perp\!\!\!\perp_{p_\perp} \mathbf{z}$. We construct a data generating process where optimal performance on every member of $\mathcal{F}$ is achieved only by uncorrelating representations that do not satisfy joint independence.

**Theorem 2.** *Define a nuisance-varying family $\mathcal{F} = \{p_D(\mathbf{y}, \mathbf{z}, \mathbf{x}) = p(\mathbf{y})p_D(\mathbf{z} \mid \mathbf{y})p(\mathbf{x} \mid \mathbf{y}, \mathbf{z})\}$. Let $\mathcal{R}_J = \{r(\mathbf{x}); [r(\mathbf{x}), \mathbf{y}] \perp\!\!\!\perp_{p_\perp} \mathbf{z}\}$ and $\mathcal{R}_C = \{r(\mathbf{x}); \mathbf{y} \perp\!\!\!\perp_{p_\perp} \mathbf{z} \mid r(\mathbf{x})\}$ be the set of representations that, under the nuisance-randomized distribution, satisfy joint independence and conditional independence respectively. Then there exists a nuisance-varying family $\mathcal{F}$ such that*

$$\forall p_{te} \in \mathcal{F} \qquad \max_{r \in \mathcal{R}_J} \mathtt{Perf}_{p_{te}}(r(\mathbf{x})) \leq \max_{r \in \mathcal{R}_C} \mathtt{Perf}_{p_{te}}(r(\mathbf{x})), \qquad (16)$$

*and $\exists p_{te} \in \mathcal{F}$ for which the inequality is strict*

$$\max_{r \in \mathcal{R}_J} \mathtt{Perf}_{p_{te}}(r(\mathbf{x})) < \max_{r \in \mathcal{R}_C} \mathtt{Perf}_{p_{te}}(r(\mathbf{x})), \qquad (17)$$

*Proof.* In this proof we will build a nuisance-varying family $\mathcal{F}$ such that $p_\perp \in \mathcal{F}$ and $\mathbf{y} \perp\!\!\!\perp_{p_\perp} \mathbf{z} \mid \mathbf{x}$. This makes $\mathbf{x}$ a maximally blocking uncorrelating representation because $r(\mathbf{x}) \perp\!\!\!\perp_{p_\perp} \mathbf{y} \mid \mathbf{x}, \mathbf{z}$. Thus it has optimal performance on every $p_{te} \in \mathcal{F}$ within the class of uncorrelating representations. We let $\mathbf{y}$ be binary, $p_\perp \in \mathcal{F}$. The structure of the rest of the proof is as follows:

1. The representation $f(\mathbf{x}) = p_\perp(\mathbf{y} = 1 \mid \mathbf{x})$ is optimal in that it performs exactly as well as $\mathbf{x}$ on every member of the family $\mathcal{F}$.

2. Any representation $T(\mathbf{x})$ that matches the performance of $\mathbf{x}$ on every $p_{te} \in \mathcal{F}$ satisfies $\mathbf{y} \perp\!\!\!\perp_{p_\perp} \mathbf{x} \mid T(\mathbf{x})$.

3. All functions $T(\mathbf{x})$ such that $\mathbf{y} \perp\!\!\!\perp_{p_\perp} \mathbf{x} \mid T(\mathbf{x})$ determine $f(\mathbf{x})$. This is shown in lemma 5.

4. We construct a family where $f(\mathbf{x}) \not\perp\!\!\!\perp_{p_\perp} \mathbf{z}$ which, by the point above, means that every optimal representation $T(\mathbf{x})$ is dependent on $\mathbf{z}$: $T(\mathbf{x}) \not\perp\!\!\!\perp_{p_\perp} \mathbf{z}$. But every representation $r \in \mathcal{R}_J$ satisfies $r(\mathbf{x}) \perp\!\!\!\perp_{p_\perp} \mathbf{z}$ and, therefore, is strictly worse in performance than $f(\mathbf{x})$ on $p_\perp$, meaning that they perform also strictly worse than $\mathbf{x}$ (because $\mathtt{Perf}_{p_{te}}(f(\mathbf{x})) = \mathtt{Perf}_{p_{te}}(\mathbf{x})$). Noting $\mathbf{x} \in \mathcal{R}_C$ completes the proof.

For 1, let $f(\mathbf{x}) = p_\perp(\mathbf{y} = 1 \mid \mathbf{x})$. However, we show here that $p_\perp(\mathbf{y} \mid f(\mathbf{x})) = p_\perp(\mathbf{y} \mid \mathbf{x})$.

$$\begin{aligned}
p_\perp(\mathbf{y} = 1 \mid f(\mathbf{x})) &= \mathbb{E}_{p_\perp(\mathbf{x} \mid f(\mathbf{x}))} p_\perp(\mathbf{y} = 1 \mid \mathbf{x}, f(\mathbf{x})) \\
&= \mathbb{E}_{p_\perp(\mathbf{x} \mid f(\mathbf{x}))} p_\perp(\mathbf{y} = 1 \mid \mathbf{x}) \\
&= \mathbb{E}_{p_\perp(\mathbf{x} \mid f(\mathbf{x}))} f(\mathbf{x}) \\
&= f(\mathbf{x}) \\
&= p_\perp(\mathbf{y} = 1 \mid \mathbf{x}) \quad (= p_\perp(\mathbf{y} \mid \mathbf{x}, f(\mathbf{x})))
\end{aligned}$$

This means $f(\mathbf{x})$ performs exactly as well as $\mathbf{x}$ on every $p_{te} \in \mathcal{F}$ and $\mathbf{x} \perp\!\!\!\perp_{p_\perp} \mathbf{y} \mid f(\mathbf{x})$.

For 2, recall $\mathtt{Perf}_{p_{te}}(r(\mathbf{x})) = -\mathbb{E}_{p_{te}(\mathbf{x})} \mathrm{KL}\left[p_{te}(\mathbf{y} \mid \mathbf{x}) \,\|\, p_\perp(\mathbf{y} \mid r(\mathbf{x}))\right]$ and note that 1 implies

$$\mathtt{Perf}_{p_\perp}(\mathbf{x}) = \mathtt{Perf}_{p_\perp}(f(\mathbf{x})) = 0.$$

Let $T(\mathbf{x})$ be any function that performs as well as $f(\mathbf{x})$ on every $p_{te} \in \mathcal{F}$. As $p_\perp \in \mathcal{F}$,

$$\begin{aligned}
0 = \mathtt{Perf}_{p_\perp}(f(\mathbf{x})) &= \mathtt{Perf}_{p_\perp}(T(\mathbf{x})) \\
&= -\mathbb{E}_{p_\perp(\mathbf{x})} \mathrm{KL}\left[p_\perp(\mathbf{y} \mid \mathbf{x}) \,\|\, p_\perp(\mathbf{y} \mid T(\mathbf{x}))\right] \\
&= -\mathbb{E}_{p_\perp(\mathbf{x})} \mathrm{KL}\left[p_\perp(\mathbf{y} \mid \mathbf{x}, T(\mathbf{x})) \,\|\, p_\perp(\mathbf{y} \mid T(\mathbf{x}))\right] \\
&= -\mathbf{I}_{p_\perp}(\mathbf{y}; \mathbf{x} \mid T(\mathbf{x})) \\
&\implies \mathbf{y} \perp\!\!\!\perp_{p_\perp} \mathbf{x} \mid T(\mathbf{x}).
\end{aligned}$$

We leave 3 to lemma 5 and show 4 here.

**The example data generating process.** We give a data generating process where $f(\mathbf{x}) = p(\mathbf{y} = 1 \mid \mathbf{x})$ is dependent on $\mathbf{z}$ : $f(\mathbf{x}) \not\perp\!\!\!\perp_{p_\perp} \mathbf{z}$. We assume $p_\perp \in \mathcal{F}$. With a binary $\mathbf{y}$ and a normal $\mathbf{z}$, let $p_\perp(\mathbf{y}, \mathbf{z}, \mathbf{x}) = p(\mathbf{y})p(\mathbf{z})p(\mathbf{x} \mid \mathbf{y}, \mathbf{z})$ be generated as follows: with $\rho : \{0, 1\} \times \{0, 1\} \to (0, 1)$, let

$$p(\mathbf{y} = 1) = 0.5, \quad \mathbf{z} \sim \mathcal{N}(0, 1), \quad p(\mathbf{b} = 1 \mid \mathbf{y} = y, \mathbf{z} = z) = \rho(y, \mathbf{1}[z \geq 0]), \quad \mathbf{x} = [\mathbf{b}, \mathbf{1}[\mathbf{z} \geq 0]].$$

We will drop the subscript in $\perp\!\!\!\perp_p$ for readability next. Throughout the next part, we use a key property of independence: $[a, b] \perp\!\!\!\perp c \iff b \perp\!\!\!\perp c \mid a, a \perp\!\!\!\perp c$.

As $\mathbf{z}$ is a standard normal random variable $\mathbf{1}[\mathbf{z} \geq 0] \perp\!\!\!\perp |\mathbf{z}| \mid \mathbf{y}$, meaning we can write $(\mathbf{y}, \mathbf{1}[\mathbf{z} \geq 0]) \perp\!\!\!\perp |\mathbf{z}|$ because $\mathbf{z}$ is generated independently of $\mathbf{y}$. Thus, as the distribution of $\mathbf{b}$ only depends on $\mathbf{1}[z \geq 0]$ and $\mathbf{y}$ due to the data generating process, it holds that $(\mathbf{b}, \mathbf{y}, \mathbf{1}[\mathbf{z} \geq 0]) \perp\!\!\!\perp |\mathbf{z}|$. Then

$$(\mathbf{b}, \mathbf{y}, \mathbf{1}[\mathbf{z} \geq 0]) \perp\!\!\!\perp |\mathbf{z}| \implies \mathbf{y} \perp\!\!\!\perp |\mathbf{z}| \mid \mathbf{b}, \mathbf{1}[\mathbf{z} \geq 0] \implies \mathbf{y} \perp\!\!\!\perp \mathbf{z} \mid \mathbf{b}, \mathbf{1}[\mathbf{z} \geq 0] \implies \mathbf{y} \perp\!\!\!\perp \mathbf{z} \mid \mathbf{x}$$

As $\mathbf{x}$ only depends on $\mathbf{1}[\mathbf{z} \geq 0]$ and $\mathbf{b}$, for readability, we define $\mathbf{a} = \mathbf{1}[\mathbf{z} \geq 0]$. Then $p_\perp(\mathbf{y}, \mathbf{a}, \mathbf{b}) = p_\perp(\mathbf{y})p_\perp(\mathbf{a})p(\mathbf{b} \mid \mathbf{y}, \mathbf{a})$, where $p(\mathbf{b} = 1 \mid \mathbf{y} = y, \mathbf{a} = a) = \rho(y, a)$ and $\mathbf{x} = [\mathbf{b}, \mathbf{a}]$.

We overload the notation for $f$: expanding $\mathbf{x} = [b, a]$, we let $f(\mathbf{x}) = f(b, a) = p(\mathbf{y} = 1 \mid \mathbf{x} = [b, a])$. We write $f(b, a)$ for different values of $b$ here,

$$f(1, a) = p(\mathbf{y} = 1 \mid \mathbf{x} = [1, a]) = p(\mathbf{y} = 1 \mid \mathbf{b} = 1, \mathbf{a} = a)$$
$$= \frac{p(\mathbf{b} = 1, \mathbf{y} = 1 \mid \mathbf{a} = a)}{p(\mathbf{b} = 1 \mid \mathbf{a} = a)}$$
$$= \frac{p(\mathbf{y} = 1)p(\mathbf{b} = 1 \mid \mathbf{y} = 1, \mathbf{a} = a)}{\sum_{y \in \{0,1\}} p(\mathbf{y} = y)p(\mathbf{b} = 1 \mid \mathbf{y} = y, \mathbf{a} = a)}$$
$$= \frac{0.5 p(\mathbf{b} = 1 \mid \mathbf{y} = 1, \mathbf{a} = a)}{0.5 \sum_{y \in \{0,1\}} (p(\mathbf{b} = 1 \mid \mathbf{y} = y, \mathbf{a} = a))}$$
$$= \frac{\rho(1, a)}{\rho(0, a) + \rho(1, a)}.$$
$$f(0, a) = p(\mathbf{y} = 1 \mid \mathbf{x} = [0, a]) = p(\mathbf{y} = 1 \mid \mathbf{b} = 0, \mathbf{a} = a)$$
$$= \frac{p(\mathbf{b} = 0, \mathbf{y} = 1 \mid \mathbf{a} = a)}{p(\mathbf{b} = 0 \mid \mathbf{a} = a)}$$
$$= \frac{p(\mathbf{y} = 1)p(\mathbf{b} = 0 \mid \mathbf{y} = 1, \mathbf{a} = a)}{\sum_{y \in \{0,1\}} p(\mathbf{y} = y)p(\mathbf{b} = 0 \mid \mathbf{y} = y, \mathbf{a} = a)}$$
$$= \frac{0.5(1 - p(\mathbf{b} = 1 \mid \mathbf{y} = 1, \mathbf{a} = a))}{0.5 \left( \sum_{y \in \{0,1\}} 1 - p(\mathbf{b} = 1 \mid \mathbf{y} = y, \mathbf{a} = a) \right)}$$
$$= \frac{1 - \rho(1, a)}{2 - \rho(0, a) - \rho(1, a)}.$$

We let $\rho(y, 1) = 0.5$ for $y \in \{0, 1\}$, $\rho(0, 0) = 0.1$, and $\rho(1, 0) = 0.9$. Then, with $a = 1$,

$$f(1, a) = \frac{\rho(1, 1)}{\rho(0, 1) + \rho(1, 1)} = \frac{0.5}{0.5 + 0.5} = 0.5, \tag{18}$$

$$f(0, a) = \frac{1 - \rho(1, 1)}{2 - \rho(0, 1) - \rho(1, a)} = \frac{1 - 0.5}{2 - 0.5 - 0.5} = 0.5, \tag{19}$$

and with $a = 0$,

$$f(1, a) = \frac{\rho(1, 0)}{\rho(0, 0) + \rho(1, 0)} = \frac{0.9}{0.1 + 0.9} = 0.9, \tag{20}$$

$$f(0, a) = \frac{1 - \rho(1, 0)}{2 - \rho(0, 0) - \rho(1, 0)} = \frac{1 - 0.9}{2 - 0.1 - 0.9} = 0.1. \tag{21}$$

Thus, the distribution $f(\mathbf{x}) \mid \mathbf{a} = a$ changes with $a$ meaning that $f(\mathbf{x}) \not\perp\!\!\!\perp \mathbf{a}$ which implies $f(\mathbf{x}) \not\perp\!\!\!\perp \mathbf{z}$ as $\mathbf{a} = \mathbf{1}[\mathbf{z} \geq a]$ is a function of $\mathbf{z}$.

Note that $f(\mathbf{x}) \not\perp\!\!\!\perp \mathbf{z}$, then $f(\mathbf{x}) \notin \mathcal{R}_J$ as $f(\mathbf{x}) \perp\!\!\!\perp \mathbf{z}$ is an implication of joint independence. Any function $T(\mathbf{x})$ that achieves the same performance as $p_\perp(\mathbf{y} \mid f(\mathbf{x}))$ (by 3 and lemma 5), determines $f(\mathbf{x})$. It follows that, $T(\mathbf{x}) \notin \mathcal{R}_J$ because

$$f(\mathbf{x}) \not\perp\!\!\!\perp_{p_\perp} \mathbf{z} \implies T(\mathbf{x}) \not\perp\!\!\!\perp_{p_\perp} \mathbf{z}.$$

So every $r \in \mathcal{R}_J$ must perform worse than $f(\mathbf{x})$, and consequently $\mathbf{x}$, on $p_\perp$. Finally, the independence $\mathbf{x} \not\perp\!\!\!\perp_{p_\perp} \mathbf{z}$ implies $\mathbf{x} \notin \mathcal{R}_J$ but $\mathbf{y} \perp\!\!\!\perp_{p_\perp} \mathbf{z} \mid \mathbf{x}$ and so $\mathbf{x} \in \mathcal{R}_C$. In this example we constructed, $\mathbf{x}$ is the maximally blocking uncorrelating representation which means that it is optimal in $\mathcal{R}_C$ on every $p_{te} \in \mathcal{F}$. As $\mathcal{R}_J$ is a subset, any $r \in \mathcal{R}_J$ can at best match the performance of $\mathbf{x}$ and we already showed that every $r \in \mathcal{R}_J$ is worse than $f(\mathbf{x})$ and consequently $\mathbf{x}$ on $p_\perp$. This completes the proof.

$\square$

**Lemma 5.** *Consider a joint distribution $p(\mathbf{y}, \mathbf{x})$ with binary $\mathbf{y}$. Assume that $p(\mathbf{x} \mid \mathbf{y} = y)$ has the same support for $y \in \{0, 1\}$. Then, for any function $T(\mathbf{x})$ such that $\mathbf{y} \perp\!\!\!\perp \mathbf{x} \mid T(\mathbf{x})$, the function $f(\mathbf{x}) = p(\mathbf{y} = 1 \mid \mathbf{x})$ is $T(\mathbf{x})$-measurable ($T(\mathbf{x})$ determines $f(\mathbf{x})$).*

*Proof.* We use the notion of sufficient statistics from estimation theory, which are defined for a family of distributions, to define the set of functions $T(\mathbf{x})$ for the joint distribution $p(\mathbf{y}, \mathbf{x})$.

**Sufficient statistics in estimation theory.** Consider a family of distributions $\mathcal{P} = \{p_\theta(\mathbf{x}); \theta \in \Omega\}$. Assume $\theta$ is discrete and that $\Omega$ is finite. A function $T(\mathbf{x})$ is a sufficient statistic of a family of distributions $\mathcal{P}$ if the conditional distribution $p_\theta(\mathbf{x} \mid T(\mathbf{x}) = t)$ does not vary with $\theta$ for (almost) any value of $t$. A minimal sufficient statistic is a sufficient statistic $M(x)$ such that for any sufficient statistic $T(X)$ $T(x) = T(x') \implies M(x) = M(x')$. Any bijective transform of $M(\mathbf{x})$ is also a minimal sufficient statistic.

The rest of the proof will follow from relying on theorem 6.12 from Lehmann and Casella (2006) which constructs a minimal sufficient statistic for a finite family of distributions $\mathcal{P} = \{p_i; i \in \{0, K-1\}\}$ as

$$M(\mathbf{x}) = \left\{ \frac{p_1(\mathbf{x})}{p_0(\mathbf{x})}, \frac{p_2(\mathbf{x})}{p_0(\mathbf{x})}, \cdots, \frac{p_{K-1}(\mathbf{x})}{p_0(\mathbf{x})} \right\}$$

**Defining the family with conditionals.** Now let the family $\mathcal{P} = \{p_y(\mathbf{x}); y \in \{0, 1\}\}$ where $p_y(\mathbf{x}) = p(\mathbf{x} \mid \mathbf{y} = y)$ which are conditionals of the joint distribution $p(\mathbf{x}, \mathbf{y})$ we were given in the theorem statement. Next, we show that the set of functions $T(\mathbf{x})$ such that $\mathbf{y} \perp\!\!\!\perp_p \mathbf{x} \mid T(\mathbf{x})$ is exactly the set of the sufficient statistics for the family $\{p(\mathbf{x} \mid \mathbf{y} = y); y \in \{0, 1\}\}$.

By definition of sufficiency where $p_y(\mathbf{x} \mid T(\mathbf{x}) = t)$ does not vary with $y$ for any value of $t$,

$$\forall t, \quad p_1(\mathbf{x} \mid T(\mathbf{x}) = t) = p_0(\mathbf{x} \mid T(\mathbf{x}) = t) \iff \forall t, \quad p(\mathbf{x} \mid T(\mathbf{x}) = t, \mathbf{y} = 1) = p(\mathbf{x} \mid T(\mathbf{x}) = t, \mathbf{y} = 0),$$

where the last statement is equivalent to the conditional independence $\mathbf{x} \perp\!\!\!\perp \mathbf{y} \mid T(\mathbf{x})$.

**Minimality of $p(\mathbf{y} = 1 \mid \mathbf{x})$.** By definition $p_y(\mathbf{x}) = p(\mathbf{x} \mid \mathbf{y} = y)$. As this family contains only two elements, the minimal sufficient statistic is

$$M(\mathbf{x}) = \frac{p_1(\mathbf{x})}{p_0(\mathbf{x})} = \frac{p(\mathbf{x} \mid \mathbf{y} = 1)}{p(\mathbf{x} \mid \mathbf{y} = 0)} = \frac{p(\mathbf{y} = 0)}{p(\mathbf{y} = 1)} \frac{p(\mathbf{y} = 1 \mid \mathbf{x})}{1 - p(\mathbf{y} = 1 \mid \mathbf{x})}.$$

Thus, $M(\mathbf{x})$ is a bijective transformation of the function $p(\mathbf{y} = 1 \mid \mathbf{x})$ (when $p(\mathbf{y} = 1 \mid \mathbf{x}) \in (0, 1)$) which in turn implies that $p(\mathbf{y} = 1 \mid \mathbf{x})$ is a minimal sufficient statistic for the family $\mathcal{P}$.

**Conclusion.** We showed that the set of functions that satisfy $\mathbf{y} \perp\!\!\!\perp \mathbf{x} \mid T(\mathbf{x})$ are sufficient statistics for the family $\mathcal{P}$. In turn, because only sufficient statistics $T(\mathbf{x})$ of the family $\mathcal{P}$ satisfy $\mathbf{y} \perp\!\!\!\perp_p \mathbf{x} \mid T(\mathbf{x})$, it follows by definition that $p(\mathbf{y} = 1 \mid \mathbf{x})$ is determined by every $T(\mathbf{x})$, completing the proof.

$\square$

## A.8 GAUSSIAN EXAMPLE OF THE INFORMATION CRITERION

**Proposition 3.** *Consider the following family of distributions $q_a$ indexed by $a \in \mathbb{R}$,*

$$\epsilon_y, \epsilon_z \sim \mathcal{N}(0, 1) \qquad \mathbf{y} \sim \mathcal{N}(0, 1) \qquad \mathbf{z} \sim \mathcal{N}(a\mathbf{y}, 1/2) \qquad \mathbf{x} = [\mathbf{y} + \epsilon_y, \mathbf{z} + \sqrt{1/2}\epsilon_z]$$

*In this family, for any $p_D = q_b(\mathbf{y} \mid \mathbf{z})$ where $\mathbf{y} \not\perp\!\!\!\perp_{p_D} \mathbf{z}$, there exists a $p'_D = q_a(\mathbf{y} \mid \mathbf{z})$ such that*

$$\left[ \mathbb{E}_{p'_D(\mathbf{x})} KL\left[ p'_D(\mathbf{y} \mid \mathbf{x}) \,\|\, p_D(\mathbf{y} \mid \mathbf{x}) \right] - \mathbf{I}_{p'_D}(\mathbf{x}; \mathbf{y}) \right] > 0.$$

*Proof.* (of proposition 3) First, write $\mathbf{z} = a\mathbf{y} + \sqrt{1/2}\delta$ where $\delta \sim \mathcal{N}(0,1)$. Let $\epsilon = \sqrt{1/2}(\delta + \epsilon_z)$; this is a normal variable with mean 0 and variance 1. Then, write $\mathbf{x} = [\mathbf{y} + \epsilon_y, a\mathbf{y} + \epsilon]$ where $\epsilon_y, \epsilon$ are Gaussian random variables with joint distribution $q(\epsilon_y)q(\epsilon)$. Therefore, $q_a(\mathbf{y}, \mathbf{x})$ is a multivariate Gaussian distribution, with the following covariance matrix (over $\mathbf{y}, \mathbf{x}_1, \mathbf{x}_2$):

$$\Sigma = \begin{pmatrix} 1 & 1 & a \\ 1 & 2 & a \\ a & a & a^2+1 \end{pmatrix} \implies \Sigma_{1,2} = [1, a], \quad \Sigma_{2,2}^{-1} = \frac{1}{a^2+2}\begin{pmatrix} a^2+1 & -a \\ -a & 2 \end{pmatrix},$$

The conditional mean and variance are:

$$\mathbb{E}_{q_a}[\mathbf{y} \mid \mathbf{x} = x] = \Sigma_{1,2}\Sigma_{2,2}^{-1}\mathbf{x} = \frac{1}{a^2+2}[1, a]\mathbf{x}$$

$$\sigma^2_{q_a}(\mathbf{y} \mid \mathbf{x}) = \Sigma_{1,1} - \Sigma_{1,2}\Sigma_{2,2}^{-1}\Sigma_{2,1} = 1 - \frac{a^2+1}{a^2+2} = \frac{1}{a^2+2}.$$

Rewrite the quantity in the theorem statement as a single expression:

$$\begin{aligned}
\mathbb{E}_{q_a(\mathbf{x})}&\mathrm{KL}\left[q_a(\mathbf{y} \mid \mathbf{x}) \parallel q_b(\mathbf{y} \mid \mathbf{x})\right] - \mathbf{I}_{q_a}(\mathbf{x}; \mathbf{y}) \\
&= \mathbb{E}_{q_a(\mathbf{x})}\mathrm{KL}\left[q_a(\mathbf{y} \mid \mathbf{x}) \parallel q_b(\mathbf{y} \mid \mathbf{x})\right] - \mathbb{E}_{q_a(\mathbf{x})}\mathrm{KL}\left[q_a(\mathbf{y} \mid \mathbf{x}) \parallel q(\mathbf{y})\right]. \\
&= \mathbb{E}_{q_a(\mathbf{x},\mathbf{y})}\log\frac{q_a(\mathbf{y} \mid \mathbf{x})}{q_b(\mathbf{y} \mid \mathbf{x})} - \mathbb{E}_{q_a(\mathbf{x},\mathbf{y})}\log\frac{q_a(\mathbf{y} \mid \mathbf{x})}{q(\mathbf{y})}. \\
&= \mathbb{E}_{q_a(\mathbf{x},\mathbf{y})}\left(\log q(\mathbf{y}) - \log q_b(\mathbf{y} \mid \mathbf{x})\right).
\end{aligned} \tag{22}$$

Expand $(\log q(\mathbf{y}) - \log q_b(\mathbf{y} \mid \mathbf{x}))$ in terms of quantities that vary with $\mathbf{y}, \mathbf{x}$ and those that do not:

$$\begin{aligned}
\log q(\mathbf{y} = y) - \log q_b(\mathbf{y} = y \mid \mathbf{x}) &= -\frac{\mathbf{y}^2}{2} - \log\sqrt{2\pi} + \frac{(\mathbf{y} - \mathbb{E}_{q_b}[\mathbf{y} \mid \mathbf{x}])^2}{2\sigma^2_{q_b}(\mathbf{y} \mid \mathbf{x})} + \log\sqrt{2\pi\sigma^2_{q_b}(\mathbf{y} \mid \mathbf{x})} \\
&= -\frac{\mathbf{y}^2}{2} - \log\sqrt{2\pi} + (b^2+2)\frac{\left(\mathbf{y} - \frac{1}{b^2+2}[1, b]\mathbf{x}\right)^2}{2} + \log\sqrt{\frac{2\pi}{b^2+2}} \\
&= -\frac{\mathbf{y}^2}{2} + (b^2+2)\frac{\left(\mathbf{y} - \frac{1}{b^2+2}[1, b]\mathbf{x}\right)^2}{2} + \log\sqrt{\frac{1}{b^2+2}}
\end{aligned}$$

As only the first two terms vary with $\mathbf{y}, \mathbf{x}$, compute the expectations $\mathbb{E}_{q_a}$ over these:

$$\begin{aligned}
\mathbb{E}_{q_a(\mathbf{x})q_a(\mathbf{y} \mid \mathbf{x})}&\left(-\frac{\mathbf{y}^2}{2} + (b^2+2)\frac{\left(\mathbf{y} - \frac{1}{b^2+2}[1, b]\mathbf{x}\right)^2}{2}\right) = \mathbb{E}_{q(\mathbf{y})}\left(-\frac{\mathbf{y}^2}{2}\right) + (b^2+2)\mathbb{E}_{q(\mathbf{y})q_a(\mathbf{x} \mid \mathbf{y})}\frac{\left(\mathbf{y} - \frac{1}{b^2+2}[1, b]\mathbf{x}\right)^2}{2} \\
&= -\frac{1}{2} + (b^2+2)\mathbb{E}_{q(\mathbf{y})q_a(\mathbf{x} \mid \mathbf{y})}\frac{\left((b^2+2)\mathbf{y} - [1, b]\mathbf{x}\right)^2}{2(b^2+2)^2} \\
&= -\frac{1}{2} + \mathbb{E}_{q(\mathbf{y})q_a(\mathbf{x} \mid \mathbf{y})}\frac{\left((b^2+2)\mathbf{y} - [1, b]\mathbf{x}\right)^2}{2(b^2+2)} \\
&= -\frac{1}{2} + \mathbb{E}_{q(\mathbf{y})q(\epsilon_y)q(\epsilon)}\frac{\left((b^2+2)\mathbf{y} - \mathbf{y} - \epsilon_y - ab\mathbf{y} - b\epsilon\right)^2}{2(b^2+2)} \\
&= -\frac{1}{2} + \mathbb{E}_{q(\mathbf{y})q(\epsilon_y)q(\epsilon)}\frac{\left((b^2+1-ab)\mathbf{y} - \epsilon_y - b\epsilon\right)^2}{2(b^2+2)} \\
&= -\frac{1}{2} + \frac{\mathrm{var}\left((b^2+1-ab)\mathbf{y}\right) + \mathrm{var}(\epsilon_y) + \mathrm{var}(b\epsilon)}{2(b^2+2)} \\
&= -\frac{1}{2} + \frac{(b^2+1-ab)^2\mathrm{var}(\mathbf{y}) + \mathrm{var}(\epsilon_y) + b^2\mathrm{var}(\epsilon)}{2(b^2+2)} \\
&= -\frac{1}{2} + \frac{(b^2+1-ab)^2 + 1 + b^2}{2(b^2+2)}
\end{aligned}$$

$$= \frac{(b^2 + 1 - ab)^2 - 1}{2(b^2 + 2)}$$

The proof follows for any $a$ such that

$$\frac{(b^2 + 1 - ab)^2 - 1}{2(b^2 + 2)} + \log \sqrt{1/b^2 + 2} = \frac{(b^2 + 1 - ab)^2 - 1}{2(b^2 + 2)} - \frac{1}{2} \log (b^2 + 2) > 0$$

Let $a = b + \frac{1+\nu}{b}$ for some scalar $\nu$. Then, if $|\nu| > 1 + (b^2 + 2) \log(b^2 + 2)$,

$$\frac{(b^2 + 1 - ab)^2 - 1}{2(b^2 + 2)} - \frac{1}{2} \log (b^2 + 2) = \frac{\nu^2 - 1}{2(b^2 + 2)} - \frac{1}{2} \log (b^2 + 2) > 0.$$

$\square$

### A.9 EXAMPLE WHERE $p_{tr}(\mathbf{y} \mid \mathbf{x}), p_{\perp}(\mathbf{y} \mid \mathbf{x})$ PERFORM WORSE THAN $p(\mathbf{y})$ UNDER $p_{te}$

In this section, we motivate nuisance-randomization and the uncorrelating property. Consider the following data generating process for a family $\{q_a\}_{a \in \mathbb{R}}$ and fixed positive scalar $\sigma^2$:

$$\mathbf{y} \sim \mathcal{N}(0,1) \quad \mathbf{z} \sim \mathcal{N}(a\mathbf{y}, 0.5) \quad \mathbf{x} = \left[\mathbf{x}_1 \sim \mathcal{N}(\mathbf{y} - \mathbf{z}, \sigma^2 - 0.5), \mathbf{x}_2 \sim \mathcal{N}(\mathbf{y} + \mathbf{z}, 0.5)\right]. \quad (23)$$

Letting $\sigma^2 = 2$ recovers the example in eq. (2). We keep $\sigma^2$ for ease of readability. We first derive the performance of $p(\mathbf{y}) = q_b(\mathbf{y}) = q(\mathbf{y})$ relative to $q_b(\mathbf{y} \mid \mathbf{x})$ under $q_a$.

**Performance of $q(\mathbf{y})$ relative to $q_b(\mathbf{y} \mid \mathbf{x})$ under $q_a$** Rewrite $\mathbf{x} = \left[(1 - a) * \mathbf{y} + \sqrt{\sigma^2}\epsilon_1, (1 + a) * \mathbf{y} + \epsilon_2\right]$, where $\epsilon_1, \epsilon_2 \sim \mathcal{N}(0,1)$. Let the joint distribution over $\mathbf{y}, \epsilon_1, \epsilon_2$ be $q(\mathbf{y})q(\epsilon_1)q(\epsilon_2)$. Then, $q_a(\mathbf{y}, \mathbf{x})$ is a multivariate Gaussian distribution with the following covariance matrix (over $\mathbf{y}, \mathbf{x}_1, \mathbf{x}_2$):

$$\Sigma = \begin{pmatrix} 1 & (1-a) & (1+a) \\ (1-a) & (1-a)^2 + \sigma^2 & (1-a^2) \\ (1+a) & (1-a^2) & (1+a)^2 + 1 \end{pmatrix},$$

$$\implies \Sigma_{1,2} = [1 - a, 1 + a], \quad \Sigma_{2,2}^{-1} = \frac{1}{\sigma^2(1+a)^2 + (1-a)^2 + \sigma^2} \begin{pmatrix} (1+a)^2 + 1 & -(1-a^2) \\ -(1-a^2) & (1-a)^2 + \sigma^2 \end{pmatrix},$$

$$\implies \mathbb{E}_{q_a}[\mathbf{y} \mid \mathbf{x} = \mathbf{x}] = \Sigma_{1,2}\Sigma_{2,2}^{-1}\mathbf{x} = \frac{1}{\sigma^2(1+a)^2 + (1-a)^2 + \sigma^2}[(1-a), \sigma^2(1+a)]\mathbf{x}$$

$$\implies \sigma_{q_a}^2(\mathbf{y} \mid \mathbf{x}) = \Sigma_{1,1} - \Sigma_{1,2}\Sigma_{2,2}^{-1}\Sigma_{2,1} = 1 - \frac{\sigma^2(1+a)^2 + (1-a)^2}{\sigma^2(1+a)^2 + (1-a)^2 + \sigma^2} = \frac{\sigma^2}{\sigma^2(1+a)^2 + (1-a)^2 + \sigma^2}.$$

The performance of $q(\mathbf{y})$ (recall Perf is negative **KL**) relative to $q_b(\mathbf{y} \mid \mathbf{x})$ on $q_a(\mathbf{y}, \mathbf{x})$ can be written as

$$\mathbb{E}_{q_a(\mathbf{x})}\text{KL}\left[q_a(\mathbf{y} \mid \mathbf{x}) \parallel q_b(\mathbf{y} \mid \mathbf{x})\right] - \mathbb{E}_{q_a(\mathbf{x})}\text{KL}\left[q_a(\mathbf{y} \mid \mathbf{x}) \parallel q(\mathbf{y})\right] = \mathbb{E}_{q_a(\mathbf{x},\mathbf{y})}\left(\log q(\mathbf{y}) - \log q_b(\mathbf{y} \mid \mathbf{x})\right).$$

Expand $(\log q(\mathbf{y}) - \log q_b(\mathbf{y} \mid \mathbf{x}))$ in terms that vary with $\mathbf{y}, \mathbf{x}$ and those that do not:

$$\log q(\mathbf{y} = \mathbf{y}) - \log q_b(\mathbf{y} = \mathbf{y} \mid \mathbf{x})$$

$$= -\frac{\mathbf{y}^2}{2} - \log \sqrt{2\pi} + \frac{(\mathbf{y} - \mathbb{E}_{q_b}[\mathbf{y} \mid \mathbf{x}])^2}{2\sigma_{q_b}^2(\mathbf{y} \mid \mathbf{x})} + \log \sqrt{2\pi\sigma_{q_b}^2(\mathbf{y} \mid \mathbf{x})}$$

$$= -\frac{\mathbf{y}^2}{2} - \log \sqrt{2\pi} + (\sigma^2(1+b)^2 + (1-b)^2 + \sigma^2)\frac{\left(\mathbf{y} - \frac{[(1-b), \sigma^2(1+b)]\mathbf{x}}{\sigma^2(1+b)^2 + (1-b)^2 + \sigma^2}\right)^2}{2\sigma^2} + \log \sqrt{2\pi\sigma_{q_b}^2(\mathbf{y} \mid \mathbf{x})}$$

$$= -\frac{\mathbf{y}^2}{2} + (\sigma^2(1+b)^2 + (1-b)^2 + \sigma^2)\frac{\left(\mathbf{y} - \frac{[(1-b), \sigma^2(1+b)]\mathbf{x}}{\sigma^2(1+b)^2 + (1-b)^2 + \sigma^2}\right)^2}{2\sigma^2} + \log \sqrt{\sigma_{q_b}^2(\mathbf{y} \mid \mathbf{x})}$$

As $\sigma_{q_b}^2(\mathbf{y} \mid \mathbf{x})$ does not vary with $\mathbf{x}, \mathbf{y}$, we will compute the expectation over the first two terms:

$$\mathbb{E}_{q_a(\mathbf{x},\mathbf{y})}\left(\log q(\mathbf{y}) - \log q_b(\mathbf{y} \mid \mathbf{x})\right)$$

$$= \mathbb{E}_{q_a(\mathbf{x})q_a(\mathbf{y} \mid \mathbf{x})} \left( -\frac{\mathbf{y}^2}{2} + (\sigma^2(1+b)^2 + (1-b)^2 + \sigma^2) \frac{\left(\mathbf{y} - \frac{1}{\sigma^2(1+b)^2+(1-b)^2+\sigma^2}[(1-b), \sigma^2(1+b)]\mathbf{x}\right)^2}{2\sigma^2} \right)$$

$$= \mathbb{E}_{q(\mathbf{y})} \left( -\frac{\mathbf{y}^2}{2} \right) + (\sigma^2(1+b)^2 + (1-b)^2 + \sigma^2)\mathbb{E}_{q(\mathbf{y})q_a(\mathbf{x} \mid \mathbf{y})} \frac{\left(\mathbf{y} - \frac{1}{\sigma^2(1+b)^2+(1-b)^2+\sigma^2}[(1-b), \sigma^2(1+b)]\mathbf{x}\right)^2}{2\sigma^2}$$

$$= -\frac{1}{2} + (\sigma^2(1+b)^2 + (1-b)^2 + \sigma^2)\mathbb{E}_{q(\mathbf{y})q_a(\mathbf{x} \mid \mathbf{y})} \frac{\left((\sigma^2(1+b)^2 + (1-b)^2 + \sigma^2)\mathbf{y} - [(1-b), \sigma^2(1+b)]\mathbf{x}\right)^2}{2\sigma^2(\sigma^2(1+b)^2 + (1-b)^2 + \sigma^2)^2}$$

$$= -\frac{1}{2} + \mathbb{E}_{q(\mathbf{y})q_a(\mathbf{x} \mid \mathbf{y})} \frac{\left((\sigma^2(1+b)^2 + (1-b)^2 + \sigma^2)\mathbf{y} - [(1-b), \sigma^2(1+b)]\mathbf{x}\right)^2}{2\sigma^2(\sigma^2(1+b)^2 + (1-b)^2 + \sigma^2)}$$

$$\left( \text{recall } \mathbf{x} = [(1-a)\mathbf{y} + \sqrt{\sigma^2}\epsilon_1, (1+a)\mathbf{y} + \epsilon_2] \right)$$

$$= -\frac{1}{2} +$$

$$\mathbb{E}_{q(\mathbf{y})q(\epsilon_y)q(\epsilon)} \frac{\left((\sigma^2(1+b)^2 + (1-b)^2 + \sigma^2)\mathbf{y} - (1-b)\sqrt{\sigma^2}\epsilon_1 - (1-a)(1-b)\mathbf{y} - \sigma^2(1+a)(1+b)\mathbf{y} - \sigma^2(1+b)\epsilon_2\right)^2}{2\sigma^2(\sigma^2(1+b)^2 + (1-b)^2 + \sigma^2)}$$

$$= -\frac{1}{2} + \mathbb{E}_{q(\mathbf{y})q(\epsilon_y)q(\epsilon)} \frac{\left(\left(\sigma^2(1+b)^2 + (1-b)^2 - \sigma^2(a+b+ab) - (1-a)(1-b)\right)\mathbf{y} - (1-b)\sqrt{\sigma^2}\epsilon_1 - \sigma^2(1+b)\epsilon_2\right)^2}{2\sigma^2(\sigma^2(1+b)^2 + (1-b)^2 + \sigma^2)}$$

$$\{\text{let } C(a,b) = \left(\sigma^2(1+b)^2 + (1-b)^2 - \sigma^2(a+b+ab) - (1-a)(1-b)\right)\}$$

$$= -\frac{1}{2} + \frac{\text{var}\left(C(a,b)\mathbf{y}\right) + \text{var}((1-b)\sqrt{\sigma^2}\epsilon_1) + \text{var}(\sigma^2(1+b)\epsilon_2)}{2\sigma^2(\sigma^2(1+b)^2 + (1-b)^2 + \sigma^2)}$$

$$= -\frac{1}{2} + \frac{(C(a,b))^2 \, \text{var}(\mathbf{y}) + \sigma^2(1-b)^2\text{var}(\epsilon_1) + (\sigma^2)^2(1+b)^2\text{var}(\epsilon_2)}{2\sigma^2(\sigma^2(1+b)^2 + (1-b)^2 + \sigma^2)}$$

$$= -\frac{1}{2} + \frac{(C(a,b))^2 + \sigma^2(1-b)^2 + (\sigma^2)^2(1+b)^2}{2\sigma^2(\sigma^2(1+b)^2 + (1-b)^2 + \sigma^2)}$$

$$= \frac{C(a,b)^2 - (\sigma^2)^2}{2\sigma^2(\sigma^2(1+b)^2 + (1-b)^2 + \sigma^2)}$$

Note that $\log \sqrt{\sigma_{q_b}^2(\mathbf{y} \mid \mathbf{x})} - \log \sqrt{\frac{2}{\sigma^2(1+b)^2+(1-b)^2+\sigma^2}} = \frac{1}{2}\log \frac{(\sigma^2(1+b)^2+(1-b)^2+\sigma^2)}{\sigma^2}$. With this, we have the ability to bound the performance of $q(\mathbf{y})$ relative to $q_b(\mathbf{y} \mid \mathbf{x})$ under $q_a$.

**The conditional $q_1(\mathbf{y} \mid \mathbf{x})$ performs worse than $q_1(\mathbf{y}) = p(\mathbf{y})$ on $q_{-1}$.** We show here that $p_{tr}(\mathbf{y} \mid \mathbf{x}) = q_1(\mathbf{y} \mid \mathbf{x})$ does worse than $q_1(\mathbf{y}) = p(\mathbf{y})$ on $p_{te} = q_{-1}$. Letting $a = -1 \implies (1+a) = 0, (1-a) = 2$ and $b = 1 \implies (1+b) = 2, (1-b) = 0$ then

$$C(a,b) = \sigma^2(1+b)^2 + (1-b)^2 - \sigma^2(a+b+ab) - (1-a)(1-b) = \sigma^2 * 2^2 + \sigma^2 = 5\sigma^2.$$

$$\implies \frac{C(a,b)^2 - (\sigma^2)^2}{2\sigma^2(\sigma^2(1+b)^2 + (1-b)^2 + \sigma^2)} - \frac{1}{2}\log \frac{\left(\sigma^2(1+b)^2 + (1-b)^2 + \sigma^2\right)}{\sigma^2}$$

$$= \frac{25(\sigma^2)^2 - (\sigma^2)^2}{2 * \sigma^2(5 * \sigma^2)} - \frac{1}{2}\log \frac{5 * \sigma^2}{\sigma^2}$$

$$= \frac{12}{5} - \frac{1}{2}\log 5 > 0.$$

**Predictive failure when $\mathbf{x}$ is not uncorrelating** Note that $q_0(\mathbf{y} \mid \mathbf{x}) = p_\perp(\mathbf{y} \mid \mathbf{x})$ corresponds to the nuisance randomized conditional of the label given the covariates. We show that $q_0(\mathbf{y} \mid \mathbf{x})$

performs worse than $p(\mathbf{y})$ on infinitely many $q_a$. Letting $b = 0$ in $C(a, b)$ helps compute the performance of $q(\mathbf{y})$ over $q_0(\mathbf{y} \mid \mathbf{x})$ for every $q_a$:

$$C(a, b) = \sigma^2(1 + b)^2 + (1 - b)^2 - \sigma^2(a + b + ab) - (1 - a)(1 - b) = \sigma^2 + 1 - a\sigma^2 - (1 - a) = \sigma^2 + a(1 - \sigma^2).$$

Then, the performance of $q(\mathbf{y})$ relative to $p_\perp(\mathbf{y} \mid \mathbf{x})$ is

$$\frac{C(a, b)^2 - (\sigma^2)^2}{2\sigma^2(\sigma^2(1 + b)^2 + (1 - b)^2 + \sigma^2)} - \frac{1}{2} \log \frac{\left(\sigma^2(1 + b)^2 + (1 - b)^2 + \sigma^2\right)}{\sigma^2}$$

$$= \frac{(a^2(1 - \sigma^2)^2 + (\sigma^2)^2 + 2\sigma^2(1 - \sigma^2)a) - (\sigma^2)^2}{2 * \sigma^2(2 * \sigma^2 + 1)} - \frac{1}{2} \log \frac{(2 * \sigma^2 + 1)}{2}$$

$$\{\text{letting } \sigma^2 = 2.\}$$

$$= \frac{a^2 - 4a}{20} - \frac{1}{2} \log \frac{5}{2}.$$

This performance difference between $q(\mathbf{y})$ and $p_\perp(\mathbf{y} \mid \mathbf{x})$ is positive for any $a > 5$ or $a < -1$. Thus, for every test distribution $q_a$ such that $a > 5$ or $a < -1$, $p_\perp(\mathbf{y} \mid \mathbf{x})$ performs worse than marginal prediction.

## B   FURTHER EXPERIMENTAL DETAILS

**Implementation details**   In section 5, the label $\mathbf{y}$ is a binary variable and, consequently, we use the Bernoulli likelihood in the predictive model and the weight model. In reweighting-NURD in practice, the estimate of the nuisance-randomized distribution $\hat{p}_\perp(\mathbf{y}, \mathbf{z}, \mathbf{x}) \propto {p_{tr}(\mathbf{y})}/{\hat{p}_{tr}(\mathbf{y} \mid \mathbf{z})} p_{tr}(\mathbf{y}, \mathbf{z}, \mathbf{x})$ with an estimated $\hat{p}_{tr}(\mathbf{y} \mid \mathbf{z})$ may have a different marginal distribution $\hat{p}_\perp(\mathbf{y}) \neq p_{tr}(\mathbf{y})$. To ensure that $p_{tr}(\mathbf{y}) = \hat{p}_\perp(\mathbf{y})$, we weight our preliminary estimate $\hat{p}_\perp$ again as $\frac{p_{tr}(\mathbf{y})}{\hat{p}_\perp(\mathbf{y})} \hat{p}_\perp(\mathbf{y}, \mathbf{z}, \mathbf{x})$.

In all the experiments, the distribution $p_\theta(\mathbf{y} \mid r_\gamma(\mathbf{x}))$ is a Bernoulli distribution parameterized by $r_\gamma$ and a scaling parameter $\theta$. In general, when the family of $p_\perp(\mathbf{y} \mid r_\gamma(\mathbf{x}))$ is unknown, learning predictive models requires a parameterization $p_\theta(\mathbf{y} \mid r_\gamma(\mathbf{x}))$. When the family is known, for example when $\mathbf{y}$ is categorical, the parameters $\theta$ are not needed because the distribution $p(\mathbf{y} \mid r_\gamma(\mathbf{x}))$ can be parameterized by the representation itself. For the critic model $p_\phi(\ell \mid \mathbf{y}, \mathbf{z}, r_\gamma(\mathbf{x}))$ in the distillation step, we use a two layer neural network with 16 hidden units and ReLU activations that takes as input $\mathbf{y}, r_\gamma(\mathbf{x})$, and a scalar representation $s_\psi(\mathbf{z})$; the critic model's parameters are $\phi, \psi$. The representation $s_\psi(\mathbf{z})$ is different in the different experiments and we give these details below.

In generative-NURD, we select models for $p(\mathbf{x} \mid \mathbf{y}, \mathbf{z})$ by using the generative objective's value on a heldout subset of the training data. For model selection, we use Gaussian likelihood in the class-conditional Gaussian experiment, binary likelihood in the colored-MNIST experiment, and squared-loss reconstruction error in the Waterbirds and chest X-ray experiments. In reweighting-NURD, we use a cross-fitting procedure where the training data is split into $K$ folds, and $K$ models are trained: for each fold, we produce weights using a model trained and validated on the other $K - 1$ folds. Hyperparameter selection for the distillation step is done using the distillation loss from eq. (6) evaluated on a heldout validation subset of the nuisance-randomized data from the first step.

In all experiments, we report results with the distillation step optimized with a fixed $\lambda = 1$ and with 1 or 2 epochs worth of critic model updates per every representation update. In setting the hyperparameter $\lambda$, a practitioner should choose the largest $\lambda$ such that optimization is still stable for different seeds and the validation loss is bounded away from that of marginal prediction. Next, we give details about each individual experiment.

**Optimal linear uncorrelating representations in Class Conditional Gaussians.**   Here, we show that $r^*(\mathbf{x}) = \mathbf{x}_1 + \mathbf{x}_2$ is the best linear uncorrelating representation in terms of performance. First let the Gaussian noises in the two coordinates of $\mathbf{x}$ (given $\mathbf{y}, \mathbf{z}$) be $\epsilon_1 \sim \mathcal{N}(0, 9)$ and $\epsilon_2 \sim \mathcal{N}(0, 0.01)$ respectively. Define $r_{u,v}(\mathbf{x}) = u\mathbf{x}_1 + v\mathbf{x}_2 = (u + v)\mathbf{y} + (v - u)\mathbf{z} + u\epsilon_1 + v\epsilon_2$. We will show that $q_0(\mathbf{z} \mid r_{u,v}(\mathbf{x}), \mathbf{y} = 1) \neq q_0(\mathbf{z} \mid r_{u,v}(\mathbf{x}), \mathbf{y} = 0)$ when $u \neq v$ and $u \neq -v$. First, $q_0(\mathbf{z}, r_{u,v}(\mathbf{x}) \mid \mathbf{y} = y)$ is a bivariate Gaussian with the following covariance matrix:

$$\Sigma_y = \begin{pmatrix} 1 & (v - u) \\ (v - u) & (v - u)^2 + 9u^2 + 0.01v^2 \end{pmatrix} \implies \Sigma_{y;1,2} = v - u, \quad \Sigma_{y;2,2}^{-1} = \frac{1}{(v - u)^2 + 9u^2 + 0.01v^2}$$

The conditional mean is:

$$\mathbb{E}_{q_a}[\mathbf{z} \mid r_{u,v}(\mathbf{x}) = r, \mathbf{y}] = \mathbb{E}[\mathbf{z} \mid \mathbf{y} = 1] + \Sigma_{1,2}\Sigma_{2,2}^{-1}(r - \mathbb{E}[r_{u,v}(\mathbf{x}) \mid \mathbf{y} = 1])$$
$$= \mathbb{E}[\mathbf{z}] + \Sigma_{1,2}\Sigma_{2,2}^{-1}(r - (u+v)\mathbf{y})$$
$$= \frac{(v-u)(r - (u+v)\mathbf{y})}{(v-u)^2 + 9u^2 + 0.01v^2}$$

which is independent of $\mathbf{y}$ if and only if $u + v = 0$ or $u - v = 0$. The conditional variance does not change with $y$ because it is determined by $\Sigma_y$ which does not change with $y$. Thus $q_0(\mathbf{z} \mid r_{u,v}(\mathbf{x}), \mathbf{y}) = q_0(\mathbf{z} \mid r_{u,v}(\mathbf{x}))$ if and only if $u = v$ or $u = -v$. When $u = v$, $r_{u,v} = 2u\mathbf{y} + \text{noise}$ and $r_{u,u} \not\perp_{q_0} \mathbf{y}$ meaning that $r_{u,u}$ helps predict $\mathbf{y}$. In contrast, when $u = -v$, $r_{u,v} = 2v\mathbf{z} + \text{noise}$ and so $r_{-v,v} \perp_{q_0} \mathbf{y} \implies q_0(\mathbf{y} \mid r_{-v,v}) = q_0(\mathbf{y})$, meaning that $q_0(\mathbf{y} \mid r_{-v,v})$ has the same performance as the marginal. However, for all $u \neq 0$, $r_{u,u} = ur_{1,1}$ is a bijective transform of $r_{1,1}$ and, therefore, $q_0(\mathbf{y} \mid r_{u,u}(\mathbf{x})) = q_0(\mathbf{y} \mid r_{1,1}(\mathbf{x}))$. Thus, within the set of linear uncorrelating representations, $r_{1,1}$ is the best because its performance dominates all others on every $p_{te} \in \mathcal{F}$.

**Implementation details for Class Conditional Gaussians.** In reweighting-NURD, the model for $p_{tr}(\mathbf{y} \mid \mathbf{z})$ is a Bernoulli distribution parameterized by a neural network with 1 hidden layer with 16 units and ReLU activations. In generative-NURD, the model for $p(\mathbf{x} \mid \mathbf{y}, \mathbf{z})$ is an isotropic Gaussian whose mean and covariance are parameterized with a neural network with one layer with 16 units and ReLU activations. We use 5 cross-fitting folds in estimating the weights in reweighting-NURD. We use weighted sampling with replacement in computing the distillation objective.

In the distillation step in both reweighting and generative-NURD, the representation $r_\gamma(\mathbf{x})$ is a neural network with one hidden layer with 16 units and ReLU activations. The critic model $p_\phi(\ell \mid \mathbf{y}, \mathbf{z}, r_\gamma(\mathbf{x}))$ consists of a neural network with 2 hidden layers with 16 units each and ReLU activations that takes as input $\mathbf{y}, r_\gamma(\mathbf{x})$, and a scalar representation $s_\psi(\mathbf{z})$ which is again a neural network with a single hidden layer of 16 units and ReLU activations.

We use cross entropy to train $\hat{p}_{tr}(\mathbf{y} \mid \mathbf{z})$, $\hat{p}_\perp(\mathbf{y} \mid r_\gamma(\mathbf{x}))$, and $p_\phi(\ell \mid \mathbf{y}, \mathbf{z}, r_\gamma(\mathbf{x}))$ using the Adam (Kingma and Ba, 2014) optimizer with a learning rate of $10^{-2}$. We optimized the model for $\hat{p}_{tr}(\mathbf{y} \mid \mathbf{z})$ for 100 epochs and the model for $\hat{p}_{tr}(\mathbf{x} \mid \mathbf{y}, \mathbf{z})$ for 300 epochs. We ran the distillation step for 150 epochs with the Adam optimizer with the default learning rate. We use a batch size of 1000 in both stages of NURD. We run the distillation step with a fixed $\lambda = 1$ and two epoch's worth of gradient steps (16) for the critic model for each gradient step of the predictive model and the representation. In this experiment, we do not re-initialize $\phi, \psi$ after a predictive model update.

**Implementation details for Colored-MNIST.** For reweighting-NURD, to use the same architecture for the representation $r_\gamma(\mathbf{x})$ and for $p_{tr}(\mathbf{y} \mid \mathbf{z})$, we construct the nuisance as a $28 \times 28$ image with each pixel being equal to the most intense pixel in the original image. In generative-NURD, we use a PixelCNN model for $p(\mathbf{x} \mid \mathbf{y}, \mathbf{z})$ with 10 masked convolutional layers each with 64 filters. The model was trained using a Bernoulli likelihood with the Adam optimizer and a fixed learning rate of $10^{-3}$ and batch size 128. We parameterize multiple models in this experiment with the following neural network: 4 convolutional layers (with $32, 64, 128, 256$ channels respectively) with ReLU activations followed by a fully connected linear layer into a single unit. Both $r_\gamma(\mathbf{x}), s_\psi(\mathbf{z})$ are parameterized by this network. Both $\hat{p}_{tr}(\mathbf{y} \mid \mathbf{z})$ in reweighting-NURD and $\hat{p}_{tr}(\mathbf{y} \mid \mathbf{x})$ for ERM are Bernoulli distributions parameterized by the network described above. We use 5 cross-fitting folds in estimating the weights in reweighting-NURD.

For the critic model $p_\phi(\ell \mid \mathbf{y}, \mathbf{z}, r_\gamma(\mathbf{x}))$ in the distillation step, we use a two-hidden-layer neural network with 16 hidden units and ReLU activations that takes as input $\mathbf{y}, r_\gamma(\mathbf{x})$, and the scalar representation $s_\psi(\mathbf{z})$; the parameters $\phi$ contain $\psi$ and the parameters for the two hidden-layer neural network. The predictive model $p_\theta(\mathbf{y} \mid r_\gamma(\mathbf{x}))$ is a Bernoulli distribution parameterized by $r_\gamma(\mathbf{x})$ multiplied by a scalar $\theta$.

We use cross entropy to train $\hat{p}_{tr}(\mathbf{y} \mid \mathbf{z})$, $\hat{p}_\perp(\mathbf{y} \mid r_\gamma(\mathbf{x}))$, and $p_\phi(\ell \mid \mathbf{y}, \mathbf{z}, r_\gamma(\mathbf{x}))$ using the Adam (Kingma and Ba, 2014) optimizer with a learning rate of $10^{-3}$. We optimized the model for $\hat{p}_{tr}(\mathbf{y} \mid \mathbf{z})$ for 20 epochs and ran the distillation step for 20 epochs with the Adam optimizer with the default learning rate. We use a batch size of 300 in both stages of NURD. We run the distillation step with a fixed $\lambda = 1$ and one epoch's worth of gradient steps (14) for the critic model for each gradient step of the predictive model and the representation. In this experiment, we do not re-initialize $\phi, \psi$ after a predictive model update.

**Implementation details for the Waterbirds experiment.** For generative-NURD, we use VQ-VAE 2 (Razavi et al., 2019) to model $p_{tr}(\mathbf{x} \mid \mathbf{y}, \mathbf{z})$. For multiple latent sizes and channels in the encoder and the decoder, we saw that the resulting generated images were insufficient to build classifiers that predict better than chance on real data. This may be because of the small training dataset that consists of only 3000 samples. The model for $\hat{p}_{\perp}(\mathbf{y} \mid r_\gamma(\mathbf{x}))$ is two feedforward layers stacked on top of the representation $r_\gamma(\mathbf{x})$. The model $\hat{p}_{tr}(\mathbf{y} \mid \mathbf{z})$ in reweighting-NURD is the same model as $\hat{p}_{\perp}(\mathbf{y} \mid r_\gamma(\mathbf{x}))$ as a function of $\mathbf{x}$. The model for $p_\phi(\ell \mid \mathbf{y}, \mathbf{z}, r_\gamma(\mathbf{x}))$ consists of a neural network with two feedforward layers that takes as input $\mathbf{y}, r_\gamma(\mathbf{x})$, and a representation $s_\psi(\mathbf{z})$. Both $r_\gamma$ and $s_\psi$ are Resnet-18 models initialized with weights pretrained on Imagenet; the parameters $\phi$ contain $\psi$ and the parameters for the two hidden-layer neural network. The model in ERM for $p_{tr}(\mathbf{y} \mid \mathbf{x})$ uses the same architecture as $\hat{p}_{\perp}(\mathbf{y} \mid r_\gamma(\mathbf{x}))$ as a function of $\mathbf{x}$. We use 5 cross-fitting folds in estimating the weights in reweighting-NURD.

We use binary cross entropy as the loss in training $\hat{p}_{tr}(\mathbf{y} \mid \mathbf{z}), p_\theta(\mathbf{y} \mid r_\gamma(\mathbf{x}))$, and $p_\phi(\ell \mid \mathbf{y}, \mathbf{z}, r_\gamma(\mathbf{x}))$ using the Adam (Kingma and Ba, 2014) optimizer with a learning rate of $10^{-3}$. We optimized the model for $\hat{p}_{tr}(\mathbf{y} \mid \mathbf{z})$ for 10 epochs and ran the distillation step for 5 epochs with the Adam optimizer with the default learning rate for all parameters except $\gamma$, which parameterizes the representation $r_\gamma$; for $\gamma$, we used 0.0005. The predictive model, the critic model, and the weight model are all optimized with a weight decay of 0.01. We use a batch size of 300 for both stages of NURD. We run the distillation step with a fixed $\lambda = 1$ and two epoch's worth of gradient steps (16) for the critic model for each gradient step of the predictive model and the representation. To prevent the critic model from overfitting, we re-initialize $\phi, \psi$ after every gradient step of the predictive model.

**Implementation details for the chest X-ray experiment.** To help with generative modeling, when creating the dataset, we remove X-ray samples from MIMIC that had all white or all black borders. We use a VQ-VAE2 (Razavi et al., 2019) to model $p(\mathbf{x} \mid \mathbf{y}, \mathbf{z})$ using code from here to both train and sample. The encoder takes the lung patch as input, and the decoder takes the quantized embeddings and the non-lung patch as input. VQ-VAE2 is hierarchical with a top latent code and a bottom latent code which are both vector-quantized and fed into the decoder to reconstruct the image. Both latents consist of $8 \times 8$ embeddings each of dimension 64. The VQ-VAE is trained for 200 epochs with Adam (Kingma and Ba, 2014) with a batch size of 256 and dropout rate of 0.1. Generating samples from the VQ-VAE2 involves sampling the top latent code conditioned on the label, followed by sampling the bottom latent code conditioned on the label and the top latent code, and passing both latent codes to the decoder. To generate from the latent codes, we build a PixelSNAIL to generate the top latent code given the label and a PixelCNN to generate the bottom latent code given the label and the top latent code. These models have 5 residual layers with 128 convolutional channels. All other details were default as in here. We train these models for 450 epochs with a batch size of 256 with a learning rate of $5 \times 10^{-5}$.

For reweighting-NURD, the model $\hat{p}_{\perp}(\mathbf{y} \mid r_\gamma(\mathbf{x}))$ is two feedforward layers stacked on top of the representation $r_\gamma(\mathbf{x})$. The model in ERM for $p_{tr}(\mathbf{y} \mid \mathbf{x})$ uses the same architecture as $\hat{p}_{\perp}(\mathbf{y} \mid r_\gamma(\mathbf{x}))$ as a function of $\mathbf{x}$. Next we use a single architecture to parameterize multiple parts in this experiment: 3 convolutional layers (each 64 channels) each followed by batch norm, and dropout with a rate of 0.5 and followed by a linear fully-connected layer into a single unit. We parameterize the two representations $r_\gamma(\mathbf{x}), s_\psi(\mathbf{z})$ with this network. To build $p_\phi(\ell \mid \mathbf{y}, \mathbf{z}, r_\gamma(\mathbf{x}))$, we stack two feedforward layers of 16 hidden units with ReLU activations on top of a concatenation of $\mathbf{y}, r_\gamma(\mathbf{x})$, and the scalar representation $s_\psi(\mathbf{z})$ as described above; the parameters $\phi$ contain $\psi$ and the parameters for the two hidden-layer neural network. We use 5 cross-fitting folds in estimating the weights in reweighting-NURD.

We use binary cross entropy as the loss in training $\hat{p}_{tr}(\mathbf{y} \mid \mathbf{z}), p_\theta(\mathbf{y} \mid r_\gamma(\mathbf{x}))$, and $p_\phi(\ell \mid \mathbf{y}, \mathbf{z}, r_\gamma(\mathbf{x}))$ using the Adam (Kingma and Ba, 2014) optimizer with a learning rate of $10^{-3}$. We use a batch size of 1000 for both stages of NURD. We optimized the model for $\hat{p}_{tr}(\mathbf{y} \mid \mathbf{z})$ for 150 epochs and ran the distillation step for 100 epochs with the Adam optimizer with the default learning rate. Only the optimization for $\hat{p}_{tr}(\mathbf{y} \mid \mathbf{z})$ has a weight decay of $1e - 2$. We run the distillation step with a fixed $\lambda = 1$ and two epoch's worth of gradient steps (20) for the critic model for each gradient step of the predictive model and the representation. To prevent the critic model from overfitting, we re-initialize $\phi, \psi$ after every gradient step of the predictive model.

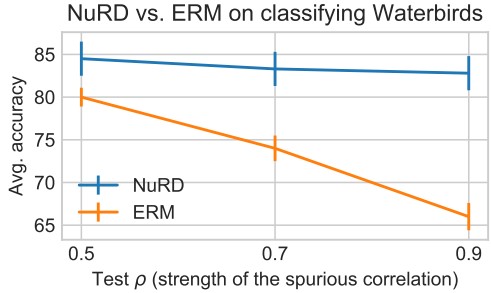 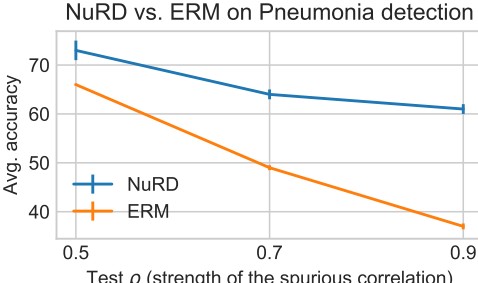

**Figure 4:** Plots of average accuracy vs. test $\rho$ for classifying Waterbirds and Pneumonia. A larger $\rho$ implies a larger difference between the nuisance-label relationship in the test data used for evaluation and the training data, which has a $\rho = 0.1$. Nuisance-randomized data corresponds to $\rho = 0.5$. Unlike NuRD, ERM's performance quickly degrades as the difference between the train and the test distributions increases.

## B.1 ADDITIONAL EXPERIMENTS

**Excluding the boundary from the images (covariates) does not improve ERM in general.** In both Waterbirds and chest X-rays, we use the easy-to-acquire border as a nuisance in NuRD. Models trained on the central (non-border) regions of the image can exploit the nuisances in the center and consequently fail to generalize when the nuisance-label relationship changes. In fact, classifiers produced by ERM on border-less images do not generalize well to the test data, producing test accuracies of $39 \pm 0.5\%$ on chest X-rays and $65 \pm 2.3\%$ on Waterbirds averaged over 10 seeds. However, as independence properties that hold for the border also hold for nuisances in the central region that are determined by the border, NuRD can use the border to control for certain nuisances in the center of the image.

**Additional experiments with NuRD.** We evaluate reweighting-NuRD further in the following ways:

1. Run NuRD on data from the training data distribution defined in section 5 and evaluate on data from test distributions $p_{te}$ with different nuisance-label relationships.
2. Train NuRD with different-sized borders as nuisances.
3. Train NuRD without a nuisance where the training and the test data have the same nuisance-label relationship; we implement this by setting the nuisance $\mathbf{z} = 0$ wherever it is passed as input in the weight model or critic model.
4. Run the distillation step with different $\lambda$.

**Different test distributions.** For this experiment, we compute the test accuracies of the models trained in the experiments in section 5 on data with different nuisance-label relationships. For both classifying Waterbirds and Pneumonia, a scalar parameter $\rho$ controls nuisance-label relationships in the data generating process. In waterbirds, $\rho = p(\mathbf{y} = waterbird \mid \text{background} = \text{land}) = p(\mathbf{y} = landbird \mid \text{background} = \text{water})$. In chest X-rays, $\rho$ corresponds to the fraction of Pneumonia cases that come from CheXpert and normal cases that come from MIMIC in the data; in this task, hospital differences are one source of nuisance-induced spurious correlations. In both tasks, $\rho = 0.1$ in the training data; as test $\rho$ increases, the nuisance-label relationship changes and becomes more different from the training data. We plot the average and standard error of accuracies aggregated over 10 seeds for different test $\rho \in \{0.5, 0.7, 0.9\}$ in fig. 4.

**Nuisance specification with different borders.** For Waterbirds, we ran NuRD with the pixels outside the central 168x168 patch (a 56 pixel border) as the nuisance. Averaged over 10 seeds, reweighting-NuRD produced a model with $81\%$ test accuracy which is similar to the accuracy achieved by NuRD using a 28-pixel border as the nuisance. In comparison, ERM achieves an accuracy of $66\%$.

**NuRD without a nuisance and no nuisance-induced spurious correlations in classifying Waterbirds.** We performed an additional experiment on classifying Waterbirds where NuRD is given a constant nuisance which is equivalent to not using the nuisance. We generated training and test

data with independence between the nuisance and the label; the nuisance-label relationship does not change between training and test. Averaged over 10 seeds, ERM achieved a test accuracy of $89 \pm 0.4\%$ and NuRD achieved a test accuracy of $88 \pm 1\%$.

**Reweighting-NuRD with different** $\lambda$. Large $\lambda$s may make optimization unstable by penalizing even small violations of joint independence. Such instabilities can lead NuRD to build predictive models that do not do better than marginal prediction, resulting in large distillation loss (log-likelihood + information loss) on the validation subset of the training data. However, a small $\lambda$ may result in NuRD learning non-uncorrelating representations which can also perform worse than chance.

We ran NuRD on the waterbirds and class-conditional Gaussians experiments with $\lambda = 5$ (instead of $\lambda = 1$ like in section 5) and found that, on a few seeds, NuRD produces models with close to $50\%$ accuracy (which is the same as majority prediction) or large information loss or both. Excluding seeds with large validation loss, reweighting-NuRD achieves an average test accuracy of $76\%$ on waterbirds and $61\%$ on class-conditional Gaussians. Annealing the $\lambda$ during training could help stabilize optimization.

In setting the hyperparameter $\lambda$ in general, a practitioner should choose the largest $\lambda$ such that optimization is still stable over different seeds and the validation loss is bounded away from that of predicting without any features.

