# OpenReview forum: "Out-of-distribution Generalization in the Presence of Nuisance-Induced Spurious Correlations"
_ICLR.cc/2022/Conference — ICLR 2022 Poster_

### Official Review · Reviewer_2F1e · 2021-10-26

**Correctness:** 3
**Technical Novelty And Significance:** 3
**Empirical Novelty And Significance:** 3
**Recommendation:** 6
**Confidence:** 3

**Details Of Ethics Concerns:**

I don't have any ethics concerns.

**Main Review:**

Strengths:

The paper proposes a method to solve an important and challenging problem of nuisance-label correlations in supervised learning. The paper provides a good theoretical analysis of the proposed method and interesting simulation results to demonstrate the applications of the proposed method. The related work section is well-structured and comprehensive including previous works using different ideas. Overall, the paper is well written and easy to follow.

Weaknesses and Suggestions:

Sections 2 and 3 only introduce the major component of the proposed method. However, it is immediately clear to me what kind of datasets we can apply the proposed method. Importantly, how do we extract the nuisance variable z out a given arbitrary labelled dataset?

Equation (1) is better to be introduced using some examples of x,y and z in the second paragraph of the introduction. For an image, why the data generating process of x is conditional on z and y. I thought both x and z are some pixels on an image.

At least in statistics, "predictor" often refers to a variable a model uses to predict the target. Here "predictor" refers to the predictive model or distribution itself. I encourage the authors to reconsider their terminology.

Measuring the performance of a predictor by KL divergence seems more reasonable for a classification task. If the label is continuous, then comparing KL divergence does not tell us exactly which model is closer to the ground truth.

In proposition 3, in the distribution p_\indep, x is generated by y and z. Then why we would have y is independent of z conditional on x? I am thinking of x is a collider of y and z in a directed graphical model.

Is it true that for any p_\indep, we can find an uncorrelated set of representations R(p_\indep)?

Why the first equality in Equation (4)?

The high-level idea of nuisance randomization distillation (NURD) is clear in the paper. I wonder if there are any distributions p(x,y,z) for which we do not want to use distillation. In order words, if there are any cases that we should optimize Equation (6) or (7) without the second mutual information term. For example, if y is a function of x, maximizing the expected log-likelihood should give the best predictor.

You mentioned quite some related methods in Section 4. Is it possible to compare with some of them, or could you provide some suggestions on when to use these related methods and when we should use  NURD?




**Summary Of The Paper:**

This paper develops a representation method called nuisance-randomized distillation (NuRD) for building predictive models with better generalizability (i.e. testing performance) without using spurious nuisance-label correlations in observed data. NuRD breaks the nuisance-label dependence and finds the most informative representation to predict the label for every distribution in the nuisance -varying family which is a set of distributions that differ only in the nuisance-label relationship. The NURD method is evaluated on several application datasets; it produces accurate predictive models on the datasets with strong spurious correlations between the label and some nuisance variables.

**Summary Of The Review:**

This paper proposes an interesting idea for an important problem. It provides a good theoretical and experimental analysis of the proposed method. I recommend accepting this paper.

---

### Official Review · Reviewer_mFHY · 2021-11-01

**Correctness:** 4
**Technical Novelty And Significance:** 3
**Empirical Novelty And Significance:** 2
**Recommendation:** 8
**Confidence:** 3

**Main Review:**

Strength
1. A great list of examples is provided that describe how spurious correlations arise.
2. Mathematic details and proofs are provided in full details for all theorems etc.
3. NuRD works extremely well on problems where nuisance can be clearly defined and its relationship with label is strong and changing a lot between train and test data.

Weaknesses
1. Too much technical details in the Introduction without adequate context, e.g. too early for “We show that the distribution of the label given the covariates under the nuisance-randomized distribution is minimax optimal for the nuisance-varying family, if the label and the nuisance are independent given the covariates” and all the text after. Hard to get intuition when the second half of Intro is just listing math properties without some high level descriptions of how to attain those properties.
2. Same for the Methods section, which repeats much of the Introduction (with some math interleaved). Would be nice to have a concrete example to illustrate ALL steps involved in NuRD and then go into the theorems, definitions, etc. As currently written, it is hard to get an integrated picture when Section 2 is just a bunch of math definitions.
3. Perhaps consider putting related work up front to provide context and state contributions with respect to existing works. Any reasons why domain adaptation and optimal transport are not mentioned?
4. Would the choice of nuisance distribution have a major effect on the results? How sensitive are results to the many parameter choices?
5. The experiments with Gaussian, color MNIST, and waterbirds seem quite extreme in terms of changing relationship between nuisance and labels. Would we see same results for less extreme scenarios? A huge problem in medicine is using genomic or imaging data to predict neurodegenerative disease status with age being nuisance. Would be great if NuRD works on this problem.
6. NuRD does not perform as well as ERM when the nuisance-label relationships do not change mainly because NuRD does not use the nuisance variables to its advantage. Is it possible to create experiments where there are no nuisance variables that can be used? Would like to see if NuRD performs as least on par with ERM in that case, i.e. not underperforming when there is no change in nuisance-label relationships.
7. For the real word problem examined in this paper, pneumonia cases are upsampled, thus might not provide realistic classification assessment, i.e. samples would be correlated. Would like to see results with controls downsampled.
8. Quantitative comparison with existing DL methods, e.g. if have z, can find representation of x that predicts y well but not z.


**Summary Of The Paper:**

This paper proposes Nuisance-Randomized Distillation (NuRD) to deal with spurious correlations induced by a changing relationship between labels and nuisance variables, where covariates are also correlated with nuisance. The paper first introduces the nuisance-randomized distribution under which labels and nuisance are independent, and shows that NuRD can find uncorrelating representations of covariates where labels remain independent from nuisance when conditioned on these covariate representations. The paper further describes how to find the optimal representation that has maximum information with labels. NuRD is evaluated on various classification tasks involving class-conditional Gaussians, colored MNIST images, waterbirds, and chest X-rays.

**Summary Of The Review:**

Overall, the paper is very rigorous in proving the mathematical guarantees of the proposed method, but the method itself is not performing that well for harder real word problems. My evaluation is based on the simplicity of experiments for scenarios where NURD worked. For the harder case (even when classification bias is introduced to the data), NURD does not perform particularly well.

Post-revision Summary
The authors have nicely addressed most of my comments. The text is still heavy on math over intuition, but improved. I am thus increasing my score.

---

### Official Review · Reviewer_Vct3 · 2021-11-02

**Correctness:** 2
**Technical Novelty And Significance:** 2
**Empirical Novelty And Significance:** 2
**Recommendation:** 5
**Confidence:** 5

**Main Review:**

I find the mathematical formalization to be a bit too much and it makes it harder to understand the core proposed ideas.

Eq 1 is a contribution of this work and it presents the problem of spurious correlations broken down into the contributions of a nuisance variable z conditioned on the label and the input features conditioned on both the label and z.

From what I understand of the NuRD method you estimate the nuisance variable using a neural network and then use this to reweight the training samples based on this. How exactly do you learn p_{tr}(z) and p_{tr}(y | z) ? As this is a core contribution of this paper please explain this very clearly.

I find the evaluations lacking. Only one type of spuriously correlated feature was used. Why not evaluate on the CelebA evaluation in Sagawa et al. (2019)? And for chest X-rays why not use the same experimental setup as (Zech et al., 2018) instead of correlating the border? Note: A version of the Zech experiment was done using all publicly available data in https://arxiv.org/abs/1910.00199

Also in the non-mnist experiments, only a single size border thickness was used. In order to explore the utility of this approach please run experiments with multiple types of spurious correlations. Possibly also varying the amount of correlation. Please show an experiment where this method doesn't work so that we may understand its limitations.


**Summary Of The Paper:**

The paper presents a formalization of spuriously correlated distractor features into a family of distributions. One goal here is to train a model to be robust against these distribution shifts. The paper presents results comparing against ERM and argues that it is unique in its problem formulation compared to other methods that make assumptions about environments or specific spurious correlations.

**Summary Of The Review:**

The proposed formalization and method are interesting. However, the presentation is not focused on the core contribution, the method is lacking detail, and the experiments do not characterize the method well.

---

### Official Review · Reviewer_S6jJ · 2021-11-03

**Correctness:** 4
**Technical Novelty And Significance:** 4
**Empirical Novelty And Significance:** 4
**Recommendation:** 8
**Confidence:** 3

**Main Review:**

**Strengths**

Nuisance-induced spurious correlations are a major issue in many applications, since they lead to models that cannot generalize well to unseen data.

This paper introduces a novel idea to address this issue (to the best of my knowledge) which is theoretically principled and well motivated. As such, I found the paper a very interesting read.

The proposed method allows to remove many of the strong assumptions that limit the application of competing ones, for example it can be used in high dimensional image classification tasks.
The experimental results show that despite highly different training and test distributions, NuRD is able to greatly reduce the influence of nuisance variables.


**Main weaknesses**

While the presented theory and results are very compelling, there are two main issues with the exposition that would greatly limit its impact in the ICLR community:
1. The theoretical exposition is unnecessarily abstract and complex.
    1. Section 2 and 3 are much heavier from the theoretical point of view than the average ICLR paper. Given the nature of the presented method, this is of course not a problem per se. However, to have an impact on the community these sections should be made much more accessible. What I find that is lacking the most, is intuition on what is being presented. One way to convey it, is to use throughout these sections a running example, e.g. the very clear cows vs penguin example of the introduction. Most of the propositions, theorems and lemmas would be way easier to understand if the authors provided intuition of what they mean if we consider x being the input image and z being its background in the running example.
    2. This is particularly important when presenting NuRD in section 3. The authors should give way more intuition to the reader using an actual example on what the randomization and distillation step do. For example, what is the practical difference of using (6) vs (7) in the cows vs penguin classification task.
    3. The introduction in page 2 is very hard to understand without more concrete examples. For being the introduction of the paper, the discussion is way too abstract for readers unfamiliar with the field.

2. From the main text, there is close to no information on how this method would be implemented in practice.
    1. After reading the main text I had an understanding of the theory, but I had no clue on what I would need to do to implement this, as well as the level of additional complexity this method introduces if I were to apply it in a standard neural network classification task. Many of the info in the appendix should be moved or at least better addressed in the main text. One possible way could be presenting how the reweighting/generative NuRD and critic model would look in a standard neural network classification task, and what optimizing (6) and (7) imply.
    2. Without the code being available, I doubt many people would know how to implement this method.


Some of the above issues might be due to the lack of space in the main text. Some ideas to gain some extra space to implement the above changes could be:
* moving lemma 1, theorem 1 and the discussion around them to the appendix. The key component for NuRD is theorem 2: lemma 1 and theorem 1 can be just seen as necessary steps to prove theorem 2. This would also allow to simplify the theoretical part of the paper.
* mostly focusing in the main text on reweighting NuRD, which seems more broadly applicable (see below). Most of the generative NuRD discussion could be covered in the appendix.
* moving parts of the related work to the appendix

**Comments/question on experiments**

1. For the more realistic experiments in 5.3 and 5.4 generative NuRD seems to struggle. Is generative NuRD feasible in practice for image classification tasks? Generating good high dimensional images remains a challenge, what if the estimate of nuisance randomized distribution is poor?
2. In real world applications assuming that the object to classify in the images are centered might not be realistic. What happens if in many images from the dataset the background z contains useful info for the classification task? How robust is NuRD to this? This could happen for example with pathologies that are in the outer region of the lungs in chest x-ray images.
3. How well does NuRD scale w.r.t standard classification? What are in the experiments the training times of ERM, generative NuRD and reweighting NuRD?
4. Section 5.1. is missing a reference to table 1




**Summary Of The Paper:**

This paper introduces a method to reduce the influence of nuisance variables in predictive tasks. This is done by fitting a nuisance-randomized distribution to the data, and finding a data representation under this distribution that makes the nuisance and the label independent. Experiments show that by using a classifier learned on this representation, the method is able to improve classification performances by limiting the impact of nuisance variables.


**Summary Of The Review:**

This paper presents a very interesting idea with competitive performances to a very relevant problem in real world applications.
The exposition of the paper however needs to be greatly simplified by giving a lot more intuition and implementation details. Due to this, in this current state I am afraid that the paper would not have the impact it deserves in the ICLR community.



=====================

*Updated review after the author's rebuttal and revision of the paper*:
I have read the paper once more, the authors did a great job improving the exposition of the paper, which is now way easier to understand and accessible to a wider audience. I have therefore increased my score to an accept.

---

### Official Review · Reviewer_frmn · 2021-11-29

**Correctness:** 2
**Technical Novelty And Significance:** 2
**Empirical Novelty And Significance:** 3
**Recommendation:** 5
**Confidence:** 4

**Main Review:**

In general, this paper studies a long-standing problem of spurious association in fitting a predictive function. The intuition of NuRD is clear. If we have data from a distribution where the nuisance variables z are randomized and do not cause the label y, the prediction will not suffer from the spurious association. Another strength of this paper is that it studies a variety of representative data sets.

The following are my major questions/comments about this paper:

1. A major concern is about the setting of the data. In the literature of this problem (e.g., Invariant causal prediction (Peters+, 2016), (Invariant Risk Minimization, Arjovsky+, 2019)), the input is O=(x,z) and we do not know which are x and which are z. For example, with a image of cow on the grass, we do not know which pixels are of cow (x) and which pixels are of grass (z). But this paper assumes we know this separation a priori. Then why not just exclude all nuisance z from the predictor and only use x as the input?

2. The paper lacks sufficient clarity (e.g, Sec. 2.). A main reason might be that the paper focus on the problem of spurious association, which is a causal problem. But it does not use a causal language and notations (e.g., do-calculus / potential outcome).

3. The simulation results might need further check. In the color MNIST example, the NuRD reaches the oracle accuracy 75%. However, this is an ideal upper bound of accuracy. Referring to [IRM, Arjovsky+, 2019], the prediction accuracy on the grey-scale images (without any spurious association) is 73%. The author may need to check/explain why NuRD outperforms this practical oracle.

4. NuRD does not compare with any methods that improves over classic ERM.

5. Some implementation details are not presented, such as how to separate x and z, how to choose the hyperparameter $\lambda$.









**Summary Of The Paper:**

The paper studies the prediction problem under spurious association. The idea is to construct a "nuisance randomized distribution" based on the observed distribution. The nuisance randomized distribution is constructed by reweighting the observed data. Then the samples from the nuisance randomized distribution are used to learn a predictive function.

**Summary Of The Review:**

In sum, the reviewer think the setting of this paper, the writing clarity, and the simulation study may need further revision and improvement before getting published.

---

### Decision · Program_Chairs · 2022-01-20

**Decision:**

Accept (Poster)

**Comment:**

The paper studies how to build predictive models that are robust to nuisance-induced spurious correlations present in the data.  It introduces nuisance-randomized distillation (NuRD), constructed by reweighting the observed data, to break the nuisance-label dependence and find the most informative representation to predict the label. Experiments on several datasets show that by using a classifier learned on this representation, NuRD is able to improve the classification performance by limiting the impact of nuisance variables. The main concerns were about the presentation and organization of the paper, which was heavily focused on the theoretical justifications but fell short in explaining the intuitions and implementation details. The revision and rebuttal have addressed some of these concerns and improved the overall exposition of the paper, based on which two reviewers raised their scores to 8. While there is still room to further improve the paper by providing more detailed discussions about the proposed algorithms, the AC considers the paper ready for publication under its current form.